# Generalized Picture Fuzzy Soft Sets and Their Application in Decision Support Systems

**Muhammad Jabir Khan** [1] , **Poom Kumam** [1,2,3,*] , **Shahzaib Ashraf** [4] **and Wiyada Kumam** [5,*]

1   KMUTT Fixed Point Research Laboratory, Room SCL 802 Fixed Point Laboratory, Science Laboratory Building, Department of Mathematics, Faculty of Science, King Mongkut's University of Technology Thonburi (KMUTT), 126 Pracha Uthit Rd., Bang Mod, Thung Khru, Bangkok 10140, Thailand; jabirkhan.uos@gmail.com
2   KMUTT-Fixed Point Theory and Applications Research Group, Theoretical and Computational Science Center (TaCS), Science Laboratory Building, Faculty of Science, King Mongkut's University of Technology Thonburi (KMUTT), 126 Pracha-Uthit Road, Bang Mod, Thrung Khru, Bangkok 10140, Thailand
3   Department of Medical Research, China Medical University Hospital, China Medical University, Taichung 40402, Taiwan
4   Department of Mathematics, Abdul wali Khan University, Mardan 23200, Pakistan; shahzaibashraf@awkum.edu.pk
5   Program in Applied Statistics, Department of Mathematics and Computer Science, Faculty of Science and Technology, Rajamangala University of Technology Thanyaburi (RMUTT), Thanyaburi, Pathumthani 12110, Thailand
*   Correspondence: poom.kumam@mail.kmutt.ac.th (P.K.); wiyada.kum@rmutt.ac.th (W.K.)

**Abstract:** In this paper, a generalized picture fuzzy soft set is proposed, which is an extension of the picture fuzzy soft sets. We investigate the basic properties of picture fuzzy soft sets and define an F-subset, M-subset, extended union, extended intersection, restricted union, restricted intersection and also prove the De Morgan's laws for picture fuzzy soft information. We investigate upper and lower substitution for both picture fuzzy sets and generalized picture fuzzy soft sets. Meanwhile, the related proofs are given in detail. Finally, we propose an algorithm to deal with generalized picture fuzzy soft information. To show the supremacy and effectiveness of the proposed technique, we illustrate a descriptive example using generalized picture fuzzy soft information. Results indicate that the proposed technique is more generalized and effective over all the existing structures of fuzzy soft sets.

**Keywords:** picture fuzzy sets; picture fuzzy soft sets; picture fuzzy Dombi weighted average operator; picture fuzzy weighted averaging operator

## 1. Introduction

This universe is loaded with qualm, imprecision, and ambiguity. In reality, the greater part of the ideas we deal contain unclear information rather than exact. Managing qualm or uncertainty is a noteworthy issue in numerous territories, for example, economics, engineering, natural science, medicinal science, and sociology. Such a large number of authors have turned out recently to have keen interest in demonstrating unclearness. Traditional speculations like fuzzy sets [1], rough sets [2] and vague sets [3] are notable and assume vital jobs in demonstrating uncertainty. In [4], an intuitionistic fuzzy set is introduced by Atanassov. In [5], Molodtsov defined soft sets which are a totally new scientific instrument for managing uncertainties.

Molodtsov soft set theory attracts many authors because it has a wide range of applications in fields of decision making, forecasting and in data analysis. Nowadays many authors try to hybridize the

soft set with different mathematical models. In [6], fuzzy soft sets are defined by Maji, which consists of both a fuzzy set and a soft set. Also, Maji [7], combine intuitionistic fuzzy set and soft set and defined an intuitionistic fuzzy soft set. Then the further extensions of soft sets like the interval-valued fuzzy soft set [8], the generalized fuzzy soft set [9], the vague soft set [10], the soft rough set [11], the trapezoidal fuzzy soft set [12], the neutrosophic soft set [13], the intuitionistic neutrosophic soft set [14], the multi-fuzzy soft set [15] and the hesitant fuzzy soft set [16] are introduced. In [17], Agarwal generalizes the notion of the intuitionist fuzzy soft set by adding a parameter which shows the director opinion about the legitimacy of the given data and called it the generalized intuitionistic fuzzy soft set. Later, the existing definition of a generalized intuitionistic fuzzy soft set is clarified and reformulated by Feng [18]. Under an uncertainty environment, these mathematical models have been successfully applied in decision making problems.

The soft matrix in a soft set, its related operations and the method of how it works for solving decision-making problems are introduced by Cagman and Enginoglu in [19]. Feng and Zhou [20], introduced the soft discernibility matrix and gave a technique to solve decision making problems.

In [21], Coung introduced the new notion which includes an extra output; the "neutral degree of membership" and called it a picture fuzzy soft set ($PFS$). It triply consists of the degree of positive membership, the degree of neutral membership, and the degree of negative membership. Picture fuzzy set attracts authors to work on it because it is directly applied to solve daily life problems. Correlation coefficients of $PFS$ and their applications in clustering analysis are introduced by Sing [22]. With the help of novel fuzzy calculations based on the $PFS$ domain time arrangement gauging and climate estimating are given by Son and Thong [23,24]. Son [25,26], defined picture fuzzy separation measures, generalized picture distance measures and picture association measures, and connected them to tackle grouping investigation under the $PFS$ condition. Son [27], proposed a novel fuzzy derivation framework on $PFS$ to enhance the inference performance of the traditional fuzzy inference system. Thong [28,29], applied a novel picture fuzzy clustering technique for complex data and particle swarm optimization. Wei [30], exhibited picture fuzzy aggregation operators method and applied it to multi attribute decision making ($MADM$) for ranking enterprise resource planning (ERP) structures. Wei [31], researched a basic leadership technique in light of the picture fuzzy weighted cross-entropy and used this to rank the choices. Based on picture fuzzy soft sets, Yang [32], defined an adjustable soft discernibility matrix and implemented it in decision making. Garg [33], contemplated aggregation operations on picture fuzzy soft set ($PFSS$) and applied it to multi criteria decision-making ($MCDM$) problems. Peng [34], determined an algorithm for $PFS$ and applied it in decision making. For more study about decision making, we refer to [35–39]. The purpose of this paper is to minimize the possible perversions in previous evaluations made by expert groups by adding an extra picture fuzzy set given by the director. Since the director is responsible for the department, he reviews and scrutinizes the general quality of evaluations made by expert groups instead of evaluating all the alternatives with respect to every characteristic. To overcome this issue, we hybridize the $PFSS$ with picture fuzzy set ($PFS$) and obtain a new mathematical model name, the generalized picture fuzzy soft set ($GPFSS$). In this paper, Sections 1 and 2 consist of an introduction and preliminaries which include the basic definition related to fuzzy sets and picture fuzzy sets. In Section 3, we define the basic properties of picture fuzzy soft sets and define the F-subset, M-subset, extended union, extended intersection, restricted union, restricted intersection and also prove the De Morgan's laws for picture fuzzy soft information. In Sections 4 and 5, we define a generalized picture fuzzy soft set and introduce its basic properties and operations. Section 6 consists of upper and lower substitution operations of generalized picture fuzzy soft sets. In Sections 7 and 8, we proposed an algorithm to deal with $GPFS$ information and the supremacy and effectiveness of the proposed technique is verified by the case study of the construction of a tower problem. Sections 9 and 10 consist of comparisons of our proposed technique with some existing techniques and the conclusion, respectively.

## 2. Preliminaries

In this section, let us briefly recall the rudiments of fuzzy sets, soft sets, fuzzy soft fuzzy sets, and picture fuzzy sets.

Zadeh [1], introduced the notion of a fuzzy set, which provides an effective framework for handling imprecision based on the view of gradualness.

**Definition 1.** *[1] A fuzzy set $\check{A}$ over the universe $\check{X}$ is defined as*

$$\check{A} = \{(f, \xi_{\check{A}}) | f \in \check{X}\},$$

*where $\xi_{\check{A}} : \check{X} \to [0,1]$, is a membership function. For each $f \in \check{X}$, $\xi_{\check{A}}(f)$ specifies the degree to which the element $f$ belongs to the fuzzy set $\check{A}$.*

In [5], Molodtsov defined the soft set which is a totally new scientific instrument for managing uncertainties from a parametrization point of view. Let $\check{X}$ be a universal set and $\check{E}$ be a parameter space. Then there is no restriction on the parameter space, that is, it might be an infinite set even if $\check{X}$ is a finite set. Mostly, parameter space consists of attributes, characteristics or properties of elements in the universal set.

**Definition 2.** *[5] Let $\check{X}$ be a universal set, $\check{E}$ a parameter space and $P(\check{X})$ the power set of $\check{X}$. A pair $(\check{F}, \check{A})$ is called a soft set over $\check{X}$, where $\check{A} \subset \check{E}$ and $\check{F}$ is a set valued mapping given by $\check{F} : \check{A} \to P(\check{X})$.*

In [6], P.K. Maji defined the fuzzy soft set, which is the hybrid model of a fuzzy set and a soft set. Since it is a hybrid model, every attribute should be characterized by a membership function, because in real life, the perception of the people is characterized by a certain degree of vagueness and imprecision. For example, to judge the beauty of women, we cannot express the information with only two crisp numbers, 0 and 1.

**Definition 3.** *[6] Let $\check{X}$ be a universal set, $\check{E}$ a parameter space and $P(\check{X})$ the set of all fuzzy subsets of $\check{X}$. A pair $(\check{F}, \check{A})$ is called a fuzzy soft set over $\check{X}$, where $\check{A} \subset \check{E}$ and $\check{F}$ is a set valued mapping given by $\check{F} : \check{A} \to P(\check{X})$.*

In [21], Coung introduced picture fuzzy sets by adding an extra membership function, namely, the degree of the neutral membership function. Basically, the model of the picture fuzzy set may be adequate in situations when we face human opinions involving more answers of the type: yes, abstain, no, refusal. Voting can be a good example of picture fuzzy set because it involves the situation of more answers of the type: yes, abstain, no, refusal.

**Definition 4.** *[21] A picture fuzzy set $(PFS)$ $\check{A}$ over the universe $\check{X}$ is defined as*

$$\check{A} = \{(f, \xi_{\check{A}}, \eta_{\check{A}}, \vartheta_{\check{A}}) | f \in \check{X}\},$$

*where $\xi_{\check{A}}(f) \in [0,1]$ is called the "degree of positive membership of $f$ in $\check{A}$", $\eta_{\check{A}}(f) \in [0,1]$ is called the "degree of neutral membership of $f$ in $\check{A}$" and $\vartheta_{\check{A}}(f) \in [0,1]$ is called the "degree of negative membership of $f$ in $\check{A}$, which satisfying the following condition $0 \leq (\xi_{\check{A}}(f) + \eta_{\check{A}}(f) + \vartheta_{\check{A}}(f)) \leq 1$, $\forall f \in \check{X}$. Then for $f \in \check{X}$, $\pi_{\check{A}}(f) = 1 - (\xi_{\check{A}}(f) + \eta_{\check{A}}(f) + \vartheta_{\check{A}}(f))$ is called the degree of refusal membership of $f$ in $\check{A}$. For PFS $(\xi_{\check{A}}(f), \eta_{\check{A}}(f), \vartheta_{\check{A}}(f))$ are said to picture fuzzy value $(PFV)$ or picture fuzzy number $(PFN)$ and each PFV can be denoted by $q = (\xi_q, \eta_q, \vartheta_q)$, where $\xi_q, \eta_q$ and $\vartheta_q \in [0,1]$, with condition that $0 \leq \xi_q + \eta_q + \vartheta_q \leq 1$.*

In [21], Coung also defined some operations as follows.

**Definition 5.** [21] *Let $\check{A}$ and $\check{B}$ be two PFSs over $\check{X}$. Then their containment, union, intersection and complement are defined as follows:*

1.  $\check{A} \subset \check{B}$, if $\xi_{\check{A}} \leq \xi_{\check{B}}$, $\eta_{\check{A}} \leq \eta_{\check{B}}$ and $\vartheta_{\check{A}} \geq \vartheta_{\check{B}}$, $\forall f \in \check{X}$,
2.  $\check{A} \cup \check{B} = \{(f, max(\xi_{\check{A}}, \xi_{\check{B}}), min(\eta_{\check{A}}, \eta_{\check{B}}), min(\vartheta_{\check{A}}, \vartheta_{\check{B}})) | \forall f \in \check{X}\}$,
3.  $\check{A} \cap \check{B} = \{(f, min(\xi_{\check{A}}, \xi_{\check{B}}), min(\eta_{\check{A}}, \eta_{\check{B}}), max(\vartheta_{\check{A}}, \vartheta_{\check{B}})) | \forall f \in \check{X}\}$,
4.  $\check{A}^c = \{(f, \vartheta_{\check{A}}, \eta_{\check{A}}, \xi_{\check{A}}) | f \in \check{X}\}$.

For comparing between two *PFVs*, the test function and accuracy function are defined.

**Definition 6.** [30] *Let $q = (\xi_q, \eta_q, \vartheta_q)$ be a picture fuzzy value (PFV). Then their score function $\check{\Theta}$ and accuracy function $\check{\omega}$ are defined as follows:*

$$\check{\Theta}(q) = \xi_q - \vartheta_q, \check{\Theta}(q) \in [-1, 1],$$

$$\check{\omega}(q) = \xi_q + \eta_q + \vartheta_q, \check{\omega}(q) \in [0, 1].$$

With the help of Definition 6, we make a comparison between two *PFVs* as follows.

**Definition 7.** [30] *Let $q$ and $p$ be two PFVs.*

1.  *If $\check{\Theta}(q) < \check{\Theta}(p)$, then $q \prec p$,*
2.  *If $\check{\Theta}(q) > \check{\Theta}(p)$, then $q \succ p$,*
3.  *If $\check{\Theta}(q) = \check{\Theta}(p)$ and $\check{\omega}(q) < \check{\omega}(p)$, then $q \prec p$,*
4.  *If $\check{\Theta}(q) = \check{\Theta}(p)$ and $\check{\omega}(q) > \check{\omega}(p)$, then $q \succ p$,*
5.  *If $\check{\Theta}(q) = \check{\Theta}(p)$ and $\check{\omega}(q) = \check{\omega}(p)$, then $q \sim p$.*

In [40], C. Jana defined the picture fuzzy Dombi weighted average (PFDWA) operator by using Dombi t-norm and Dombi t-conorm. In Section 7, we use PFDWA to aggregate the information of an alternative from parameters.

**Definition 8.** [40] *Let $q_i = (\xi_i, \eta_i, \vartheta_i)$ $(i = 1, 2, 3, ..., n)$ be PFVs. Then the PFDWA operator is a function defined by $q^n \to q$ such that*

$$PFDWA_{\check{\omega}}(q_1, q_2, ..., q_n) = \oplus_{i=1}^{n} \check{\omega}_i q_i$$

$$= \left(1 - \frac{1}{1 + \{\sum_{i=1}^{n} \check{\omega}_i (\frac{\xi_i}{1 - \xi_i})^k\}^{\frac{1}{k}}}, \frac{1}{1 + \{\sum_{i=1}^{n} \check{\omega}_i (\frac{1 - \eta_i}{\eta_i})^k\}^{\frac{1}{k}}}, \frac{1}{1 + \{\sum_{i=1}^{n} \check{\omega}_i (\frac{1 - \vartheta_i}{\vartheta_i})^k\}^{\frac{1}{k}}}\right),$$

*where $k \geq 1$ and $\check{\omega} = (\check{\omega}_1, \check{\omega}_2, ..., \check{\omega}_n)$ is the weight vector with each $\check{\omega}_i > 0$ and $\sum_{i=1}^{n} \check{\omega}_i = 1$.*

In [30], G. Wei defined the picture fuzzy weighted averaging (PFWA) operator by using arithmetic operations. In Section 7, we also use PFWA to aggregate the information of an alternative from parameters.

**Definition 9.** [30] *Let $q_i = (\xi_i, \eta_i, \vartheta_i)$ $(i = 1, 2, 3, ..., n)$ be PFVs. Then the picture fuzzy weighted averaging (PFWA) operator is a function defined $q^n \to q$ such that*

$$PFWA_{\check{\omega}}(q_1, q_2, ..., q_n) = \oplus_{i=1}^{n} \check{\omega}_i q_i$$

$$= \left(1 - \prod_{i=1}^{n} (1 - \xi_i)^{\check{\omega}_i}, \prod_{i=1}^{n} (\eta_i)^{\check{\omega}_i}, \prod_{i=1}^{n} (\vartheta_i)^{\check{\omega}_i}\right),$$

*where $\check{\omega} = (\check{\omega}_1, \check{\omega}_2, ..., \check{\omega}_n)$ is the weight vector with each $\check{\omega}_i > 0$ and $\sum_{i=1}^{n} \check{\omega}_i = 1$.*

### 3. Picture Fuzzy Soft Sets

In [32], Y. Yang defined the *PFSS*, which is a hybrid model of picture fuzzy set and soft set. With the help of *PFSS*, we can see uncertainties from a parametrization point of view in picture fuzzy environment, that is, every element (alternative) of a universal set $\check{X}$ can be viewed from different parameters (attributes).

**Definition 10.** *[32] Let $\check{X}$ be a universal set and $\check{E}$ a parameter space. Let $PF(\check{X})$ denote the set of all picture fuzzy sets of $\check{X}$. A pair $(\check{F}, \check{A})$ is called a picture fuzzy soft set (PFSS), where $\check{A} \subset \check{E}$ and $\check{F}$ is a mapping given by $\check{F} : \check{A} \to PF(\check{X})$.*

From the definition of *PFSS*, we can see that it is not a set, but it is a parametrized family of picture fuzzy subsets of $\check{X}$. For any $h \in \check{A}$, $\check{F}(h)$ is *PFS* of $\check{X}$. Clearly, $\check{F}(h)$ can be written as a picture fuzzy set such that $\check{F}(h) = \{(f, \xi_{\check{F}(h)}, \eta_{\check{F}(h)}, \vartheta_{\check{F}(h)}) | f \in \check{X}\}$, where $\xi_{\check{F}(h)}, \eta_{\check{F}(h)}$ and $\vartheta_{\check{F}(h)}$ are the positive membership, neutral membership, and negative membership functions, respectively. Y. Yang et al. [32], also defined the equality and complement of *PFSSs*.

**Definition 11.** *Let $(\check{X}, \check{E})$ be a universe space and $\check{A}, \check{B} \subseteq \check{E}$. Suppose that $L_1 = (\check{F}, \check{A})$ and $L_2 = (\check{G}, \check{B})$ be two PFSSs over $\check{X}$. Then $L_1$ is said to be picture fuzzy soft equal to $L_2$, denoted by $L_1 = L_2$, if $\check{A} = \check{B}$ and $\check{F}(h) = \check{G}(h)$ for all $h \in \check{A}$.*

**Definition 12.** *Let $(\check{X}, \check{E})$ be a universe space and $\check{A} \subseteq \check{E}$. Suppose that $L = (\check{F}, \check{A})$ is a PFSS over $\check{X}$. The complement of $L$ is defined as the PFSS $L^c = (\check{G}, \check{A})$ such that $\check{G}(h) = [\check{F}(h)]^c$ for all $h \in \check{A}$.*

**Example 1.** *Consider a PFSS $(\check{F}, \check{A})$ over $\check{X}$, where $\check{X} = \{f_1, f_2, f_3, f_4, f_5, f_6\}$ are six laptops under consideration of the decision makers to purchase and parameter space is given by $\check{E} = \{h_1, h_2, h_3, h_4, h_5\}$, where each $h_i$ stands for "battery life", "portability", "keyboard/touch pad", "cheap" and "hard drive/RAM", respectively. Let $\check{A} = \{h_1, h_2, h_3, h_5\} \subset \check{E}$ chosen by an observer. In the view of criteria "battery life", "portability", "keyboard/touch pad" and "hard drive/RAM" are the most useful characteristics for evaluation. Evaluation is made by the customer and respective results are described by the PFSS, $(\check{F}, \check{A})$, where*

$$\check{F}(h_1) = \{(0.7, 0.1, 0.1)/f_1, (0.3, 0.2, 0.4)/f_2, (0.1, 0.5, 0.3)/f_3, (0.4, 0.1, 0.3)/f_4,$$
$$(0.2, 0.5, 0.2)/f_5, (0.6, 0.1, 0.2)/f_6\},$$

$$\check{F}(h_2) = \{(0.5, 0.1, 0.3)/f_1, (0.3, 0.2, 0.4)/f_2, (0.2, 0.3, 0.4)/f_3, (0.6, 0.2, 0.2)/f_4,$$
$$(0.5, 0.2, 0.3)/f_5, (0.6, 0.1, 0.2)/f_6\},$$

$$\check{F}(h_3) = \{(0.4, 0.1, 0.5)/f_1, (0.1, 0.2, 0.5)/f_2, (0.5, 0.1, 0.3)/f_3, (0.4, 0.1, 0.5)/f_4,$$
$$(0.7, 0.1, 0.2)/f_5, (0.3, 0.2, 0.4)/f_6\},$$

$$\check{F}(h_5) = \{(0.5, 0.2, 0.2)/f_1, (0.3, 0.1, 0.4)/f_2, (0.6, 0.1, 0.2)/f_3, (0.3, 0.2, 0.3)/f_4,$$
$$(0.4, 0.1, 0.3)/f_5, (0.2, 0.1, 0.5)/f_6\}.$$

*The tabular representation of $(\check{F}, \check{A})$ is shown in the Table 1.*

**Table 1.** The picture fuzzy soft set, $PFSS = (\check{F}, \check{A})$.

| $\check{X}$ | $h_1$ | $h_2$ | $h_3$ | $h_5$ |
|---|---|---|---|---|
| $f_1$ | (0.7, 0.1, 0.1) | (0.5, 0.1, 0.3) | (0.4, 0.1, 0.5) | (0.5, 0.2, 0.2) |
| $f_2$ | (0.3, 0.2, 0.4) | (0.3, 0.2, 0.4) | (0.1, 0.2, 0.5) | (0.3, 0.1, 0.4) |
| $f_3$ | (0.1, 0.5, 0.3) | (0.2, 0.3, 0.4) | (0.5, 0.1, 0.3) | (0.6, 0.1, 0.2) |
| $f_4$ | (0.4, 0.1, 0.3) | (0.6, 0.2, 0.2) | (0.4, 0.1, 0.5) | (0.3, 0.2, 0.3) |
| $f_5$ | (0.2, 0.5, 0.2) | (0.5, 0.2, 0.3) | (0.7, 0.1, 0.2) | (0.4, 0.1, 0.3) |
| $f_6$ | (0.6, 0.1, 0.2) | (0.6, 0.1, 0.2) | (0.3, 0.2, 0.4) | (0.2, 0.1, 0.5) |

Now, we define two types of containment in *PFSSs*, namely, F-subset and M-subset which covers every aspect of containment in *PFSSs*.

**Definition 13.** *Let $(\check{X}, \check{E})$ be a universe space and $\check{A}, \check{B} \subseteq \check{E}$. Suppose that $L_1 = (\check{F}, \check{A})$ and $L_2 = (\check{G}, \check{B})$ be two PFSSs over $\check{X}$. Then $L_1$ is said to be picture fuzzy soft F-subset of $L_2$, denoted by $L_1 \subseteq_F L_2$, if $\check{A} \subseteq \check{B}$ and $\check{F}(h) \subseteq \check{G}(h)$ for all $h \in \check{A}$.*

**Definition 14.** *Let $(\check{X}, \check{E})$ be a universe space and $\check{A}, \check{B} \subseteq \check{E}$. Suppose that $L_1 = (\check{F}, \check{A})$ and $L_2 = (\check{G}, \check{B})$ be two PFSSs over $\check{X}$. Then $L_1$ is said to be picture fuzzy soft M-subset of $L_2$, denoted by $L_1 \subseteq_M L_2$, if $\check{A} \subseteq \check{B}$ and $\check{F}(h) = \check{G}(h)$ for all $h \in \check{A}$.*

Now, we define the operations of extended union and extended intersection for *PFSSs* as follows.

**Definition 15.** *Let $L_1 = (\check{F}, \check{A})$ and $L_2 = (\check{G}, \check{B})$ be two PFSSs over $\check{X}$. Then extended union of $L_1$ and $L_2$ is defined as the PFSS $(\check{H}, \check{C}) = (\check{F}, \check{A}) \cup_\epsilon (\check{G}, \check{B})$, where $\check{C} = \check{A} \cup \check{B}$ and for all $h \in \check{C}$,*

$$\check{H}(h) = \begin{cases} \check{F}(h), & \text{if } h \in \check{A} \setminus \check{B}, \\ \check{G}(h), & \text{if } h \in \check{B} \setminus \check{A}, \\ \check{F}(h) \cup \check{G}(h), & \text{if } h \in \check{A} \cap \check{B}. \end{cases}$$

**Definition 16.** *Let $L_1 = (\check{F}, \check{A})$ and $L_2 = (\check{G}, \check{B})$ be two PFSSs over $\check{X}$. Then extended intersection of $L_1$ and $L_2$ is defined as the PFSS $(\check{H}, \check{C}) = (\check{F}, \check{A}) \cap_\epsilon (\check{G}, \check{B})$, where $\check{C} = \check{A} \cup \check{B}$ and for all $h \in \check{C}$,*

$$\check{H}(h) = \begin{cases} \check{F}(h), & \text{if } h \in \check{A} \setminus \check{B}, \\ \check{G}(h), & \text{if } h \in \check{B} \setminus \check{A}, \\ \check{F}(h) \cap \check{G}(h), & \text{if } h \in \check{A} \cap \check{B}. \end{cases}$$

Next definitions of restricted union and intersection of *PFSSs* are given.

**Definition 17.** *Let $L_1 = (\check{F}, \check{A})$ and $L_2 = (\check{G}, \check{B})$ be two PFSSs over $\check{X}$ such that $\check{C} = \check{A} \cap \check{B} \neq \emptyset$. Then the restricted union of $L_1$ and $L_2$ is defined as the PFSS $(\check{H}, \check{C}) = (\check{F}, \check{A}) \cup_R (\check{G}, \check{B})$, where $\check{H}(h) = \check{F}(h) \cup \check{G}(h)$, for all $h \in \check{C}$.*

**Definition 18.** *Let $L_1 = (\check{F}, \check{A})$ and $L_2 = (\check{G}, \check{B})$ be two PFSSs over $\check{X}$ such that $\check{C} = \check{A} \cap \check{B} \neq \emptyset$. Then the restricted intersection of $L_1$ and $L_2$ is defined as the PFSS $(\check{H}, \check{C}) = (\check{F}, \check{A}) \cap_R (\check{G}, \check{B})$, where $\check{H}(h) = \check{F}(h) \cap \check{G}(h)$, for all $h \in \check{C}$.*

**Remark 1.** *The notion of extended union and extended intersection become identical with the restricted union and restricted intersection, respectively, when we have the same set of parameters for two PFSSs.*

Now, we prove basic properties of the extended union, extended intersection, restricted union and restricted intersection in *PFSSs*.

**Theorem 1.** *Let $L = (\breve{F}, \breve{A})$ be a PFSS. Then the following properties hold:*

1.  $L \cup_\epsilon L = L \cup_R L = L,$
2.  $L \cap_\epsilon L = L \cap_R L = L.$

**Proof.** Straightforward.  □

**Theorem 2.** *Let $L_1 = (\breve{F}, \breve{A})$ and $L_1 = (\breve{G}, \breve{B})$ be two PFSSs. Then the following properties hold:*

1.  $L_1 \cup_\epsilon L_2 = L_2 \cup_\epsilon L_1,$
2.  $L_1 \cap_\epsilon L_2 = L_2 \cap_\epsilon L_1,$
3.  $L_1 \cup_R L_2 = L_2 \cup_R L_1;$
4.  $L_1 \cap_R L_2 = L_2 \cap_R L_1.$

**Proof.** Straightforward.  □

Now, we prove De Morgan's laws for extended union and extended intersection in *PFSSs*.

**Theorem 3.** *Let $(\breve{F}, \breve{A})$ and $(\breve{G}, \breve{B})$ be two PFSSs over $\breve{X}$. Then*

1.  $[(\breve{F}, \breve{A}) \cup_\epsilon (\breve{G}, \breve{B})]^c = (\breve{F}, \breve{A})^c \cap_\epsilon (\breve{G}, \breve{B})^c,$ *for all $\breve{A}, \breve{B} \subseteq \breve{E},$*
2.  $[(\breve{F}, \breve{A}) \cap_\epsilon (\breve{G}, \breve{B})]^c = (\breve{F}, \breve{A})^c \cup_\epsilon (\breve{G}, \breve{B})^c,$ *for all $\breve{A}, \breve{B} \subseteq \breve{E}.$*

**Proof.** L.H.S.
Let

$$(\breve{H}, \breve{C}) = (\breve{F}, \breve{A}) \cup_\epsilon (\breve{G}, \breve{B}), \text{ where } \breve{C} = \breve{A} \cup \breve{B}.$$

Then

$$(\breve{H}, \breve{C})^c = [(\breve{F}, \breve{A}) \cup_\epsilon (\breve{G}, \breve{B})]^c.$$

For all $h \in \breve{C} = \breve{A} \cup \breve{B}$, $\breve{H}(h)$ has the form

$$\breve{H}(h) = \begin{cases} \breve{F}(h), & \text{if } h \in \breve{A} \setminus \breve{B}, \\ \breve{G}(h), & \text{if } h \in \breve{B} \setminus \breve{A}, \\ \breve{F}(h) \cup \breve{G}(h), & \text{if } h \in \breve{A} \cap \breve{B}; \end{cases}$$

then

$$\breve{H}^c(h) = \begin{cases} \breve{F}^c(h), & \text{if } h \in \breve{A} \setminus \breve{B}, \\ \breve{G}^c(h), & \text{if } h \in \breve{B} \setminus \breve{A}, \\ [\breve{F}(h) \cup \breve{G}(h)]^c, & \text{if } h \in \breve{A} \cap \breve{B}. \end{cases}$$

Since De Morgan's laws hold in *PFSs* [21], therefore,

$$\breve{H}^c(h) = \begin{cases} \breve{F}^c(h), & \text{if } h \in \breve{A} \setminus \breve{B}, \\ \breve{G}^c(h), & \text{if } h \in \breve{B} \setminus \breve{A}, \\ \breve{F}^c(h) \cap \breve{G}^c(h), & \text{if } h \in \breve{A} \cap \breve{B}. \end{cases} \quad \in (\breve{H}, \breve{C})^c$$

R.H.S.
Let

$$(\breve{I}, \breve{C}) = (\breve{F}, \breve{A})^c \cap_\epsilon (\breve{G}, \breve{B})^c, \text{ where } \breve{C} = \breve{A} \cup \breve{B}.$$

For all $h \in \check{C} = \check{A} \cup \check{B}$ and $\check{I}(h) \in (\check{I}, \check{C})$ has the form

$$\check{I}(h) = \begin{cases} \check{F}^c(h), & \text{if } h \in \check{A} \setminus \check{B}, \\ \check{G}^c(h), & \text{if } h \in \check{B} \setminus \check{A}, \\ \check{F}^c(h) \cap \check{G}^c(h), & \text{if } h \in \check{A} \cap \check{B}. \end{cases} \in (\check{H}, \check{C})^c$$

From the above calculations and the fact that both sides have the same set of parameters, therefore, L.H.S=R.H.S. □

Similarly, we can prove the second part of the theorem.

Also, we can prove De Morgan's laws for restricted union and restricted intersection in *PFSSs*.

**Theorem 4.** *Let* $(\check{F}, \check{A})$ *and* $(\check{G}, \check{B})$ *be two PFSSs over* $\check{X}$. *Then*

1.　$[(\check{F}, \check{A}) \cup_R (\check{G}, \check{B})]^c = (\check{F}, \check{A})^c \cap_R (\check{G}, \check{B})^c$, *for all* $\check{A}, \check{B} \subseteq \check{E}$,
2.　$[(\check{F}, \check{A}) \cap_R (\check{G}, \check{B})]^c = (\check{F}, \check{A})^c \cup_R (\check{G}, \check{B})^c$, *for all* $\check{A}, \check{B} \subseteq \check{E}$.

**Proof.** L.H.S.

Let $\check{C} = \check{A} \cap \check{B} \neq \varnothing$, and

$$(\check{H}, \check{C}) = (\check{F}, \check{A}) \cup_R (\check{G}, \check{B}), \text{ where } \check{C} = \check{A} \cap \check{B}.$$

Then

$$(\check{H}, \check{C})^c = [(\check{F}, \check{A}) \cup_R (\check{G}, \check{B})]^c.$$

For all $h \in \check{C} = \check{A} \cap \check{B}$, $\check{H}(h) \in (\check{H}, \check{C})$ has the form $\check{H}(h) = \check{F}(h) \cup \check{G}(h)$. Then $\check{H}^c(h) \in (\check{H}, \check{C})^c$ has the form

$$\check{H}^c(h) = \check{F}^c(h) \cap \check{G}^c(h),$$

since De Morgan's laws hold in *PFSs* [21].

R.H.S.

Let

$$(\check{I}, \check{C}) = (\check{F}, \check{A})^c \cap_R (\check{G}, \check{B})^c, \text{ where } \check{C} = \check{A} \cap \check{B}.$$

Then for all $h \in \check{C} = \check{A} \cap \check{B}$, we have $\check{I}(h) = \check{F}^c(h) \cap \check{G}^c(h)$. From the above calculations and the fact that both sides have the same set of parameters, therefore, the proof is complete. □

Similarly, we can easily prove the second part of the theorem.

## 4. Generalized Picture Fuzzy Soft Sets

In this section, we define a generalized picture fuzzy soft set (*GPFSS*), which is an extension of picture fuzzy soft set (*PFSS*). *GPFSS* is a hybrid model of picture fuzzy soft set and picture fuzzy set. In *GPFSS*, we have an extra output in the form of picture fuzzy set in $\check{A}$. Actually, the concept of picture fuzzy soft set arising from picture fuzzy set is generalized by adding a parameter reflecting a director or moderator's opinion about the validity of the information provided. The resulting generalized picture fuzzy soft set finds a special role in the decision-making applications. Keeping in mind the idea of decision making, if there is a committee for taking an important decision, the committee evaluates the given alternatives according to the given criteria (attributes) in the form of *PFSS*. To minimize the possible perversion in previous evaluation made by committee director reviews and scrutinizes the general quality of evaluation made by the committee and give their opinion in the form of *PFS*.

**Definition 19.** *Let* $\check{X}$ *be a universal set,* $\check{A} \subset \check{E}$ *a parametric set and* $\P(\check{A})$ *the set of all picture fuzzy subsets of* $\check{A}$. *By a generalized picture fuzzy soft set we mean a triple* $(\check{F}, \check{A}, \rho)$, *where* $(\check{F}, A)$ *is a PFSS over* $\check{X}$ *and* $\rho : \check{A} \to \P(\check{A})$ *is a PFS in* $\check{A}$.

Keeping the idea of decision-making in mind, we called $(\check{F}, \check{A})$ the basic picture fuzzy soft sets ($BPFSS$) and $\rho$ is called the parametric picture fuzzy set ($PPFS$) of the generalized picture fuzzy soft set $GPFSS$ $(\check{F}, \check{A}, \rho)$. Clearly, from the definition we can see that $\rho$ is a picture fuzzy set in $\check{A}$ and can be written as $\rho = \{(h, \xi_\rho, \eta_\rho, \vartheta_\rho) | \forall\, h \in \check{A}\}$, which is different from the Definition 10, where we have picture fuzzy set over universal set $\check{X}$, which can be written as $\check{F}(h) = \{(f, \xi_{\check{F}(h)}, \eta_{\check{F}(h)}, \vartheta_{\check{F}(h)}) | f \in \check{X}\}$.

We denote the collection of all generalized picture fuzzy soft set over $\check{X}$ is $GPFSS^{\check{E}}(\check{X})$, where $\check{E}$ is a parametric space and $GPFSS_{\check{A}}(\check{X})$ for the fixed parametric space $\check{A} \subset \check{E}$.

**Example 2.** *Consider a GPFSS $(\check{F}, \check{A}, \rho)$ over $\check{X}$, where $(\check{F}, \check{A})$ be PFSS in Example 1, and $\rho$ a PPFS which is given by*

$$\rho = \{(0.3, 0.3, 0.2)/h_1, (0.5, 0.2, 0.3)/h_2, (0.2, 0.2, 0.5)/h_3, (0.7, 0.1, 0.2)/h_5\},$$

*which describes an additional opinion of a moderator on the general quality of work done for evaluating alternatives on the basis of given criteria (attributes). All the data about laptops is summarized in terms of GPFSS $(\check{F}, \check{A}, \rho)$, whose tabular representation shown in Table 2.*

**Table 2.** The generalized picture fuzzy soft set, $GPFSS = (\check{F}, \check{A}, \rho)$.

| $\check{X}$ | $h_1$ | $h_2$ | $h_3$ | $h_5$ |
|---|---|---|---|---|
| $f_1$ | $(0.7, 0.1, 0.1)$ | $(0.5, 0.1, 0.3)$ | $(0.4, 0.1, 0.5)$ | $(0.5, 0.2, 0.2)$ |
| $f_2$ | $(0.3, 0.2, 0.4)$ | $(0.3, 0.2, 0.4)$ | $(0.1, 0.2, 0.5)$ | $(0.3, 0.1, 0.4)$ |
| $f_3$ | $(0.1, 0.5, 0.3)$ | $(0.2, 0.3, 0.4)$ | $(0.5, 0.1, 0.3)$ | $(0.6, 0.1, 0.2)$ |
| $f_4$ | $(0.4, 0.1, 0.3)$ | $(0.6, 0.2, 0.2)$ | $(0.4, 0.1, 0.5)$ | $(0.3, 0.2, 0.3)$ |
| $f_5$ | $(0.2, 0.5, 0.2)$ | $(0.5, 0.2, 0.3)$ | $(0.7, 0.1, 0.2)$ | $(0.4, 0.1, 0.3)$ |
| $f_6$ | $(0.6, 0.1, 0.2)$ | $(0.6, 0.1, 0.2)$ | $(0.3, 0.2, 0.4)$ | $(0.2, 0.1, 0.5)$ |
| $\rho$ | $(0.3, 0.3, 0.2)$ | $(0.5, 0.2, 0.3)$ | $(0.2, 0.2, 0.5)$ | $(0.7, 0.1, 0.2)$ |

Based on our new definitions of F-subset and M-subset in *PFSSs*, we extend these definitions to *GPFSSs*, which covers every aspect of containment in *GPFSSs*.

**Definition 20.** *Let $(\check{X}, \check{E})$ be a universe space and $\check{A}, \check{B} \subseteq \check{E}$. Suppose that $\Gamma_1 = (\check{F}, \check{A}, \rho)$ and $\Gamma_2 = (\check{G}, \check{B}, \sigma)$ be two GPFSSs over $\check{X}$. The $\Gamma_1$ is said to be generalized picture fuzzy soft F-subset of $\Gamma_2$, denoted by $\Gamma_1 \sqsubseteq_F \Gamma_2$, if the following conditions satisfied:*

1. $(\check{F}, \check{A}) \subseteq_F (\check{G}, \check{B})$;
2. $\xi_\rho(h) \leq \xi_\sigma(h), \eta_\rho(h) \leq \eta_\sigma(h)$ and $\vartheta_\rho(h) \geq \vartheta_\sigma(h)$, for all $h \in \check{A}$.

**Definition 21.** *Let $(\check{X}, \check{E})$ be a universe space and $\check{A}, \check{B} \subseteq \check{E}$. Suppose that $\Gamma_1 = (\check{F}, \check{A}, \rho)$ and $\Gamma_2 = (\check{G}, \check{B}, \sigma)$ be two GPFSSs over $\check{X}$. The $\Gamma_1$ is said to be generalized picture fuzzy soft M-subset of $\Gamma_2$, denoted by $\Gamma_1 \sqsubseteq_M \Gamma_2$, if the following conditions satisfied:*

1. $(\check{F}, \check{A}) \subseteq_M (\check{G}, \check{B})$;
2. $\xi_\rho(h) \leq \xi_\sigma(h), \eta_\rho(h) \leq \eta_\sigma(h)$ and $\vartheta_\rho(h) \geq \vartheta_\sigma(h)$, for all $h \in \check{A}$.

The equality and complement of *GPFSSs* are defined as follows.

**Definition 22.** *The two GPFSSs $\Gamma_1 = (\check{F}, \check{A}, \rho)$ and $\Gamma_2 = (\check{G}, \check{B}, \sigma)$ are said to be generalized picture fuzzy soft equal and denoted by $\Gamma_1 = \Gamma_2$, if $\check{A} = \check{B}$, $(\check{F}, \check{A}) = (\check{G}, \check{A})$ and $\rho = \sigma$.*

**Definition 23.** *Suppose $\Gamma_1 = (\check{F}, \check{A}, \rho)$ be a GPFSS over $\check{X}$. The complement of $\Gamma_1$ is defined as the GPFSS $(\check{F}, \check{A}, \rho)^c = (\check{G}, \check{A}, \sigma)$ where $(\check{G}, \check{A})$ is the complement of the BPFSS $(\check{F}, \check{A})$ and $\sigma$ is the complement of PPFS $\rho$, respectively.*

## 5. Basic Operations of Generalized Picture Fuzzy Soft Sets

In this section, we define the operations of extended union, extended intersection, restricted union and restricted intersection for *GPFSSs*, which are helpful in the decision support system. Also, we prove some basic properties and De Morgan's laws for these operations.

Now, we define the operations of extended union and extended intersection for *GPFSSs* as follows.

**Definition 24.** *Let* $\Gamma_1 = (\check{F}, \check{A}, \rho)$ *and* $\Gamma_2 = (\check{G}, \check{B}, \sigma)$ *be two GPFSSs over* $\check{X}$. *Then extended union is denoted by* $(\check{H}, \check{C}, \tau) = (\check{F}, \check{A}, \rho) \sqcup_\epsilon (\check{G}, \check{B}, \sigma)$ *and defined as*

- $(\check{H}, \check{C}) = (\check{F}, \check{A}) \cup_\epsilon (\check{G}, \check{B})$, *where* $\check{C} = \check{A} \cup \check{B}$.
- *For all* $h \in \check{C} = \check{A} \cup \check{B}$,

$$\xi_\tau(h) = \begin{cases} \xi_\rho(h), & \text{if } h \in \check{A} \setminus \check{B}, \\ \xi_\sigma(h), & \text{if } h \in \check{B} \setminus \check{A}, \\ \max\{\xi_\rho(h), \xi_\sigma(h)\}, & \text{if } h \in \check{A} \cap \check{B}, \end{cases}$$

- *for all* $h \in \check{C} = \check{A} \cup \check{B}$,

$$\eta_\tau(h) = \begin{cases} \eta_\rho(h), & \text{if } h \in \check{A} \setminus \check{B}, \\ \eta_\sigma(h), & \text{if } h \in \check{B} \setminus \check{A}, \\ \min\{\eta_\rho(h), \eta_\sigma(h)\}, & \text{if } h \in \check{A} \cap \check{B}, \end{cases}$$

- *for all* $h \in \check{C} = \check{A} \cup \check{B}$,

$$\vartheta_\tau(h) = \begin{cases} \vartheta_\rho(h), & \text{if } h \in \check{A} \setminus \check{B}, \\ \vartheta_\sigma(h), & \text{if } h \in \check{B} \setminus \check{A}, \\ \min\{\vartheta_\rho(h), \eta_\sigma(h)\}, & \text{if } h \in \check{A} \cap \check{B}. \end{cases}$$

**Definition 25.** *Let* $\Gamma_1 = (\check{F}, \check{A}, \rho)$ *and* $\Gamma_2 = (\check{G}, \check{B}, \sigma)$ *be two GPFSSs over* $\check{X}$. *Then extended intersection is denoted by* $(\check{H}, \check{C}, \tau) = (\check{F}, \check{A}, \rho) \sqcap_\epsilon (\check{G}, \check{B}, \sigma)$ *and defined as*

- $(\check{H}, \check{C}) = (\check{F}, \check{A}) \cap_\epsilon (\check{G}, \check{B})$, *where* $\check{C} = \check{A} \cup \check{B}$.
- *For all* $h \in \check{C} = \check{A} \cup \check{B}$,

$$\xi_\tau(h) = \begin{cases} \xi_\rho(h), & \text{if } h \in \check{A} \setminus \check{B}, \\ \xi_\sigma(h), & \text{if } h \in \check{B} \setminus \check{A}, \\ \min\{\xi_\rho(h), \xi_\sigma(h)\}, & \text{if } h \in \check{A} \cap \check{B}, \end{cases}$$

- *for all* $h \in \check{C} = \check{A} \cup \check{B}$,

$$\eta_\tau(h) = \begin{cases} \eta_\rho(h), & \text{if } h \in \check{A} \setminus \check{B}, \\ \eta_\sigma(h), & \text{if } h \in \check{B} \setminus \check{A}, \\ \min\{\eta_\rho(h), \eta_\sigma(h)\}, & \text{if } h \in \check{A} \cap \check{B}, \end{cases}$$

- *for all $h \in \check{C} = \check{A} \cup \check{B}$,*

$$\vartheta_\tau(h) = \begin{cases} \vartheta_\rho(h), & \text{if } h \in \check{A} \setminus \check{B}, \\ \vartheta_\sigma(h), & \text{if } h \in \check{B} \setminus \check{A}, \\ max\{\vartheta_\rho(h), \vartheta_\sigma(h)\}, & \text{if } h \in \check{A} \cap \check{B}. \end{cases}$$

The operations of restricted union and restricted intersection are defined for *GPFSSs* as follows.

**Definition 26.** *Let $\Gamma_1 = (\check{F}, \check{A}, \rho)$ and $\Gamma_2 = (\check{G}, \check{B}, \sigma)$ be two GPFSSs over $\check{X}$ such that $\check{C} = \check{A} \cap \check{B} \neq \emptyset$. Then restricted union of $\Gamma_1$ and $\Gamma_1$ is defined as the GPFSS*

$$(\check{H}, \check{C}, \tau) = (\check{F}, \check{A}, \rho) \sqcup_R (\check{G}, \check{B}, \sigma)$$

*such that*

- $(\check{H}, \check{C}) = (\check{F}, \check{A}) \cup_R (\check{G}, \check{B})$;
- *for all $h \in \check{H}, \xi_\tau(h) = max\{\xi_\rho(h), \xi_\sigma(h)\}, \eta_\tau(h) = min\{\eta_\rho(h), \eta_\sigma(h)\}$ and*

$$\vartheta_\tau = min\{\vartheta_\rho(h), \vartheta_\sigma(h)\}.$$

**Definition 27.** *Let $\Gamma_1 = (\check{F}, \check{A}, \rho)$ and $\Gamma_2 = (\check{G}, \check{B}, \sigma)$ be two GPFSSs over $\check{X}$ such that $\check{C} = \check{A} \cap \check{B} \neq \emptyset$. Then the restricted intersection of $\Gamma_1$ and $\Gamma_1$ is defined as the GPFSS*

$$(\check{H}, \check{C}, \tau) = (\check{F}, \check{A}, \rho) \sqcap_R (\check{G}, \check{B}, \sigma)$$

*such that*

- $(\check{H}, \check{C}) = (\check{F}, \check{A}) \cap_R (\check{G}, \check{B})$;
- *for all $h \in \check{H}, \xi_\tau(h) = min\{\xi_\rho(h), \xi_\sigma(h)\}, \eta_\tau(h) = min\{\eta_\rho(h), \eta_\sigma(h)\}$ and*

$$\vartheta_\tau = max\{\vartheta_\rho(h), \vartheta_\sigma(h)\}.$$

**Example 3.** *A person wants to go to spend holidays and has four cities as alternatives $\check{X} = \{f_1, f_2, f_3, f_4\}$. He has different characteristics in his mind, that is, attributes $\check{E} = \{h_1, h_2, h_3, h_4, h_5\}$, where each $h_i$ stands for "picnic place", 'shopping place", "cheap", "distance from the house", and "weather conditions", respectively. Let $\check{A} = \{h_1, h_3, h_4\} \subset \check{E}$ and $\check{B} = \{h_1, h_2, h_3, h_5\} \subset \check{E}$ chosen by an observer. Now, according to attributes, the evaluation is made by the person and respective results are describe as a PFSSs $(\check{F}, \check{A})$ and $(\check{G}, \check{B})$, where*

$$\check{F}(h_1) = \{(0.4, 0.2, 0.1)/f_1, (0.5, 0.2, 0.2)/f_2, (0.3, 0.3, 0.3)/f_3, (0.2, 0.2, 0.5)/f_4\},$$

$$\check{F}(h_3) = \{(0.6, 0.1, 0.1)/f_1, (0.6, 0.1, 0.2)/f_2, (0.2, 0.3, 0.4)/f_3, (0.3, 0.5, 0.1)/f_4\},$$

$$\check{F}(h_4) = \{(0.6, 0.2, 0.1)/f_1, (0.5, 0.2, 0.2)/f_2, (0.4, 0.4, 0.2)/f_3, (0.2, 0.4, 0.2)/f_4\}.$$

*In addition, $\rho$ is the PPFS which is given by*

$$\rho = \{(0.7, 0.1, 0.1)/h_1, (0.5, 0.3, 0.1)/h_3, (0.3, 0.2, 0.4)/h_4\},$$

*which complete the GPFSS $(\check{F}, \check{A}, \rho)$, whose tabular representation is shown in Table 3. Also,*

$$\check{G}(h_1) = \{(0.3, 0.5, 0.1)/f_1, (0.4, 0.3, 0.2)/f_2, (0.2, 0.3, 0.4)/f_3, (0.1, 0.5, 0.3)/f_4\},$$

$$\check{G}(h_2) = \{(0.4, 0.5, 0.1)/f_1, (0.2, 0.1, 0.5)/f_2, (0.3, 0.4, 0.2)/f_3, (0.1, 0.6, 0.1)/f_4\},$$

$$\check{G}(h_3) = \{(0.7, 0.1, 0.1)/f_1, (0.2, 0.2, 0.5)/f_2, (0.4, 0.1, 0.3)/f_3, (0.1, 0.7, 0.1)/f_4\},$$

$$\check{G}(h_5) = \{(0.3, 0.2, 0.4)/f_1, (0.6, 0.1, 0.1)/f_2, (0.5, 0.1, 0.3)/f_3, (0.2, 0.2, 0.5)/f_4, \}.$$

*In addition, $\sigma$ is the PPFS which is given by*

$$\sigma = \{(0.5, 0.2, 0.3)/h_1, (0.3, 0.4, 0.2)/h_2, (0.2, 0.3, 0.4)/h_3, (0.1, 0.4, 0.4)/h_5\},$$

*which completes the GPFSS $(\check{G}, \check{B}, \sigma)$, whose tabular representation is shown in Table 4.*

**Table 3.** The *GPFSS* $(\check{F}, \check{A}, \rho)$.

| $\check{X}$ | $h_1$ | $h_3$ | $h_4$ |
|---|---|---|---|
| $f_1$ | (0.4, 0.2, 0.1) | (0.6, 0.1, 0.1) | (0.6, 0.2, 0.1) |
| $f_2$ | (0.5, 0.2, 0.2) | (0.6, 0.1, 0.2) | (0.5, 0.2, 0.2) |
| $f_3$ | (0.3, 0.3, 0.3) | (0.2, 0.3, 0.4) | (0.4, 0.4, 0.2) |
| $f_4$ | (0.2, 0.2, 0.5) | (0.3, 0.5, 0.1) | (0.2, 0.4, 0.2) |
| $\rho$ | (0.7, 0.1, 0.1) | (0.5, 0.3, 0.1) | (0.3, 0.2, 0.4) |

**Table 4.** The *GPFSS* $(\check{G}, \check{B}, \sigma)$.

| $\check{X}$ | $h_1$ | $h_2$ | $h_3$ | $h_5$ |
|---|---|---|---|---|
| $f_1$ | (0.3, 0.5, 0.1) | (0.4, 0.5, 0.1) | (0.7, 0.1, 0.1) | (0.3, 0.2, 0.4) |
| $f_2$ | (0.4, 0.3, 0.2) | (0.2, 0.1, 0.5) | (0.2, 0.2, 0.5) | (0.6, 0.1, 0.1) |
| $f_3$ | (0.2, 0.3, 0.4) | (0.3, 0.4, 0.2) | (0.4, 0.1, 0.3) | (0.5, 0.1, 0.3) |
| $f_4$ | (0.1, 0.5, 0.3) | (0.1, 0.6, 0.1) | (0.1, 0.7, 0.1) | (0.2, 0.2, 0.5) |
| $\rho$ | (0.5, 0.2, 0.3) | (0.3, 0.4, 0.2) | (0.2, 0.3, 0.4) | (0.1, 0.4, 0.4) |

*First, we consider the extended union*

$$(\check{H}_1, \check{A} \cup \check{B}, \tau_1) = (\check{F}, \check{A}, \rho) \sqcup_\epsilon (\check{G}, \check{B}, \sigma).$$

*By calculation,*

$$\tau_1 = \{(0.7, 0.1, 0.1)/h_1, (0.3, 0.4, 0.2)/h_2, (0.5, 0.3, 0.1)/h_3, (0.3, 0.2, 0.4)/h_4,$$

$$(0.1, 0.4, 0.4)/h_5\}.$$

*Moreover, we have*

$$\check{H}_1(h_1) = \{(0.4, 0.2, 0.1)/f_1, (0.5, 0.2, 0.2)/f_2, (0.3, 0.3, 0.3)/f_3, (0.2, 0.2, 0.3)/f_4\},$$

$$\check{H}_1(h_2) = \{(0.4, 0.5, 0.1)/f_1, (0.2, 0.1, 0.5)/f_2, (0.3, 0.4, 0.2)/f_3, (0.1, 0.6, 0.1)/f_4\},$$

$$\check{H}_1(h_3) = \{(0.7, 0.1, 0.1)/f_1, (0.6, 0.1, 0.2)/f_2, (0.4, 0.1, 0.3)/f_3, (0.3, 0.5, 0.1)/f_4\},$$

$$\check{H}_1(h_4) = \{(0.6, 0.2, 0.1)/f_1, (0.5, 0.2, 0.2)/f_2, (0.4, 0.4, 0.2)/f_3, (0.2, 0.4, 0.2)/f_4\},$$

$$\check{H}_1(h_5) = \{(0.3, 0.2, 0.4)/f_1, (0.6, 0.1, 0.1)/f_2, (0.5, 0.1, 0.3)/f_3, (0.2, 0.2, 0.5)/f_4, \}.$$

*Similarly, we can find extended intersection as follows*

$$(\check{H}_2, \check{A} \cup \check{B}, \tau_2) = (\check{F}, \check{A}, \rho) \sqcap_\epsilon (\check{G}, \check{B}, \sigma),$$

*which we calculate in Example 4. In addition, the restricted union is obtained as follows*

$$(\check{H}_3, \check{A} \cap \check{B}, \tau_3) = (\check{F}, \check{A}, \rho) \sqcup_R (\check{G}, \check{B}, \sigma),$$

*and the restricted intersection*

$$(\check{H}_4, \check{A} \cap \check{B}, \tau_4) = (\check{F}, \check{A}, \rho) \sqcap_R (\check{G}, \check{B}, \sigma),$$

*which are given in Tables 5 and 6.*

**Table 5.** The *GPFSS* $(\check{H}_3, \check{A} \cap \check{B}, \tau_3) = (\check{F}, \check{A}, \rho) \sqcup_R (\check{G}, \check{B}, \sigma)$.

| $\check{X}$ | $h_1$ | $h_3$ |
|---|---|---|
| $f_1$ | (0.4, 0.2, 0.1) | (0.7, 0.1, 0.1) |
| $f_2$ | (0.5, 0.2, 0.2) | (0.6, 0.2, 0.2) |
| $f_3$ | (0.3, 0.3, 0.3) | (0.4, 0.1, 0.3) |
| $f_4$ | (0.2, 0.2, 0.3) | (0.3, 0.5, 0.1) |
| $\tau_3$ | (0.7, 0.1, 0.1) | (0.5, 0.3, 0.1) |

**Table 6.** The *GPFSS* $(\check{H}_4, \check{A} \cap \check{B}, \tau_4) = (\check{F}, \check{A}, \rho) \sqcap_R (\check{G}, \check{B}, \sigma)$.

| $\check{X}$ | $h_1$ | $h_3$ |
|---|---|---|
| $f_1$ | (0.3, 0.2, 0.1) | (0.6, 0.1, 0.1) |
| $f_2$ | (0.4, 0.2, 0.2) | (0.2, 0.1, 0.5) |
| $f_3$ | (0.2, 0.3, 0.4) | (0.2, 0.1, 0.4) |
| $f_4$ | (0.1, 0.2, 0.5) | (0.1, 0.5, 0.1) |
| $\tau_4$ | (0.5, 0.1, 0.3) | (0.2, 0.3, 0.4) |

**Remark 2.** *The notion of extended union and extended intersection become identical with the restricted union and restricted intersection, respectively, when we have the same set of parameters for two GPFSSs.*

Now, we prove some properties of the extended union, extended intersection, restricted union, and restricted intersection.

**Theorem 5.** *Let* $\Gamma = (\check{F}, \check{A}, \rho)$ *be GPFSS. Then we have*

1.  $\Gamma \sqcup_\epsilon \Gamma = \Gamma \sqcup_R \Gamma = \Gamma$;
2.  $\Gamma \sqcap_\epsilon \Gamma = \Gamma \sqcap_R \Gamma = \Gamma$.

**Proof.** Straightforward.  □

**Theorem 6.** *Let* $\Gamma_1 = (\check{F}, \check{A}, \rho)$ *and* $\Gamma_2 = (\check{G}, \check{B}, \sigma)$ *be two GPFSSs. Then we have*

1.  $\Gamma_1 \sqcup_\epsilon \Gamma_2 = \Gamma_2 \sqcup_\epsilon \Gamma_1$;
2.  $\Gamma_1 \sqcap_\epsilon \Gamma_2 = \Gamma_2 \sqcap_\epsilon \Gamma_1$;
3.  $\Gamma_1 \sqcup_R \Gamma_2 = \Gamma_2 \sqcup_R \Gamma_1$;
4.  $\Gamma_1 \sqcap_R \Gamma_2 = \Gamma_2 \sqcap_R \Gamma_1$.

**Proof.** Straightforward.  □

Now, we prove De Morgan's laws for the extended union, extended intersection, restricted union, and restricted intersection.

**Theorem 7.** *Let* $(\check{F}, \check{A}, \rho)$ *and* $(\check{G}, \check{B}, \sigma)$ *be two GPFSSs over* $\check{X}$. *Then we have*

1.  $[(\check{F}, \check{A}, \rho) \sqcup_\epsilon (\check{G}, \check{B}, \sigma)]^c = (\check{F}, \check{A}, \rho)^c \sqcap_\epsilon (\check{G}, \check{B}, \sigma)^c$, *for all* $\check{A}, \check{B} \subseteq \check{E}$;
2.  $[(\check{F}, \check{A}, \rho) \sqcap_\epsilon (\check{G}, \check{B}, \sigma)]^c = (\check{F}, \check{A}, \rho)^c \sqcup_\epsilon (\check{G}, \check{B}, \sigma)^c$, *for all* $\check{A}, \check{B} \subseteq \check{E}$.

**Proof.** From Theorem 3 and the fact that De Morgans laws hold in *PFSs* [21], we can easily complete our proof.  □

Also, we can prove De Morgan's laws for restricted union and restricted intersection in *GPFSSs*.

**Theorem 8.** *Let $(\check{F}, \check{A}, \rho)$ and $(\check{G}, \check{B}, \sigma)$ be two GPFSSs over $\check{X}$. Then we have*

1.  $[(\check{F}, \check{A}, \rho) \sqcup_R (\check{G}, \check{B}, \sigma)]^c = (\check{F}, \check{A}, \rho)^c \sqcap_R (\check{G}, \check{B}, \sigma)^c$, *for all $\check{A}, \check{B} \subseteq \check{E}$;*
2.  $[(\check{F}, \check{A}, \rho) \sqcap_R (\check{G}, \check{B}, \sigma)]^c = (\check{F}, \check{A}, \rho)^c \sqcup_R (\check{G}, \check{B}, \sigma)^c$, *for all $\check{A}, \check{B} \subseteq \check{E}$.*

**Proof.** From Theorem 4 and the fact that De Morgans laws hold in *PFSs* [21], we can easily complete our proof. □

## 6. Substitution Operations of Generalized Picture Fuzzy Soft Sets

In this section, we define upper and lower substitutions for *GPFSSs* and prove some important results related to it.

**Definition 28.** *Let $\rho = \{(f, \xi_\rho(f), \eta_\rho(f), \vartheta_\rho(f)) | f \in \check{X}\}$ be a PFS. Then*

$$\{(f, max(\xi_\rho(f), \vartheta_\rho(f)), \eta_\rho(f), min(\xi_\rho(f), \vartheta_\rho(f)) | f \in \check{X}\}$$

*is called the upper substitution picture fuzzy set of $\rho$ and denoted by $U(\rho)$.*

**Definition 29.** *Let $\rho = \{(f, \xi_\rho(f), \eta_\rho(f), \vartheta_\rho(f)) | f \in \check{X}\}$ be a PFS. Then*

$$\{(f, min(\xi_\rho(f), \vartheta_\rho(f)), \eta_\rho(f), max(\xi_\rho(f), \vartheta_\rho(f)) | f \in \check{X}\}$$

*is called the lower substitution picture fuzzy set of $\rho$ and denoted by $L(\rho)$.*

**Theorem 9.** *Let $\rho$ be a PFS in $\check{X}$. Then $U(\rho)$ and $L(\rho)$ are also PFSs over $\check{X}$ such that $L(\rho) \subseteq \rho \subseteq U(\rho)$.*

**Proof.** Let $\rho = \{(f, \xi_\rho(f), \eta_\rho(f), \vartheta_\rho(f)) | f \in \check{X}\}$ be a *PFS*. Then $0 \le \xi_\rho(f) + \eta_\rho(f) + \vartheta_\rho(f) \le 1$.
If

$$\sigma = U(\rho) = \{(f, \xi_\sigma(f), \eta_\sigma(f), \vartheta_\sigma(f)) | f \in \check{X}\},$$

then

$$\xi_\sigma(f) = max(\xi_\rho(f), \vartheta_\rho(f)) \text{ and } \vartheta_\sigma(f) = min(\xi_\rho(f), \vartheta_\rho(f)).$$

If $\xi_\sigma(f) = \xi_\rho(f)$, then $\vartheta_\sigma(f) = \vartheta_\rho(f)$, therefore, $0 \le \xi_\sigma(f) + \eta_\sigma(f) + \vartheta_\sigma(f) \le 1$ and hence $\sigma = U(\rho)$ is a *PFS*. Similarly, we can prove $L(\rho)$ is also a *PFS*. In addition, from the definition it is clear that

$$min(\xi_\rho(f), \vartheta_\rho(f)) \le \xi_\rho(f) \text{ and } \vartheta_\rho(f) \le max(\xi_\rho(f), \vartheta_\rho(f)),$$

which implies that $L(\rho) \subseteq \rho$. Similarly, we have $\rho \subseteq U(\rho)$. □

**Theorem 10.** *Let $\rho$ be a PFS in $\check{X}$. Then we have*

1.  $[U(\rho)]^c = L(\rho),$
2.  $[L(\rho)]^c = U(\rho),$
3.  $U(\rho^c) = U(\rho),$
4.  $L(\rho^c) = L(\rho).$

**Proof.** Straightforward. □

**Theorem 11.** *Let $\rho$ be a PFS in $\check{X}$. Then we have*

1.  $U(\rho) = \rho \cup \rho^c,$
2.  $L(\rho) = \rho \cap \rho^c.$

**Proof.** Straightforward. □

**Theorem 12.** *Let $\rho$ and $\sigma$ be two PFSs in $\check{X}$. Then we have*

1.  $U(\rho \cup \sigma) = (U(\rho) \cup \sigma) \cap (\rho \cup U(\sigma))$,
2.  $U(\rho \cap \sigma) = (U(\rho) \cup \sigma^c) \cap (\rho^c \cup U(\sigma))$.

**Proof.** By Theorem 11, we have

$$U(\rho \cup \sigma) = (\rho \cup \sigma) \cup (\rho \cup \sigma)^c.$$

Since De Morgan's laws hold in *PFSs* [21], therefore,

$$\begin{aligned}
U(\rho \cup \sigma) &= (\rho \cup \sigma) \cup (\rho^c \cap \sigma^c) \\
&= (\rho \cup \sigma \cup \rho^c) \cap (\rho \cup \sigma \cup \sigma^c) \\
&= (\rho \cup \rho^c \cup \sigma) \cap (\rho \cup \sigma \cup \sigma^c) \\
&= (U(\rho) \cup \sigma) \cap (\rho \cup U(\sigma)).
\end{aligned}$$

This completes our first proof.
Similarly, for second proof, we have

$$U(\rho \cap \sigma) = (\rho \cap \sigma) \cup (\rho \cap \sigma)^c.$$

Since De Morgan's laws hold in *PFSs* [21], therefore,

$$\begin{aligned}
U(\rho \cap \sigma) &= (\rho \cap \sigma) \cup (\rho^c \cup \sigma^c) \\
&= (\rho \cup \rho^c \cup \sigma^c) \cap (\sigma \cup \rho^c \cup \sigma^c) \\
&= (\rho \cup \rho^c \cup \sigma) \cap (\rho \cup \sigma \cup \sigma^c) \\
&= (U(\rho) \cup \sigma^c) \cap (\rho^c \cup U(\sigma)).
\end{aligned}$$

This completes our second proof. □

**Theorem 13.** *Let $\rho$ and $\sigma$ be two PFSs in $\check{X}$. Then we have*

1.  $L(\rho \cup \sigma) = (L(\rho) \cap \sigma^c) \cup (\rho^c \cap L(\sigma))$,
2.  $L(\rho \cap \sigma) = (L(\rho) \cap \sigma) \cup (\rho \cap L(\sigma))$.

**Proof.** The proof is similar to the proof of Theorem 12. □

**Definition 30.** *Let $\Gamma = (\check{F}, \check{A}, \rho)$ be a GPFSS. Then $\check{U}(\Gamma) = (\check{G}, \check{A}, \sigma)$ is called an upper substitution GPFSS of $\Gamma$ if the following conditions hold:*

1.  $\sigma = U(\rho)$,
2.  $\check{G}(h) = U(\check{F}(h))$, *for all $h \in \check{A}$.*

**Definition 31.** *Let $\Gamma = (\check{F}, \check{A}, \rho)$ be a GPFSS. Then $\check{L}(\Gamma) = (\check{G}, \check{A}, \sigma)$ is called lower substitution GPFSS of $\Gamma$ if the following conditions hold:*

1.  $\sigma = L(\rho)$,
2.  $\check{G}(h) = L(\check{F}(h))$, *for all $h \in \check{A}$.*

**Theorem 14.** *Let $\Gamma = (\check{F}, \check{A}, \rho)$ be a GPFSS in $\check{X}$. Then $\check{U}(\check{F}, \check{A}, \rho)$ and $\check{L}(\check{F}, \check{A}, \rho)$ are also GPFSSs over $\check{X}$ such that $\check{L}(\check{F}, \check{A}, \rho) \subseteq_F \Gamma \subseteq_F \check{U}(\check{F}, \check{A}, \rho)$.*

**Proof.** This theorem follows directly from Definitions 20, 30, 31 and Theorem 9. □

Now, we prove an important theorem of this section.

**Theorem 15.** *Let* $\Gamma = (\check{F}, \check{A}, \rho)$ *be a GPFSS over* $\check{X}$. *Then we have*

1.  $\Gamma \sqcup_\epsilon \Gamma^c = \Gamma \sqcup_R \Gamma^c = \check{U}(\Gamma)$;
2.  $\Gamma \sqcap_\epsilon \Gamma^c = \Gamma \sqcap_R \Gamma^c = \check{L}(\Gamma)$.

**Proof.** Since $\Gamma$ and $\Gamma^c$ have the same set of parameters $\check{A}$, therefore, by Remark 2, we have

$$\Gamma \sqcup_\epsilon \Gamma^c = \Gamma \sqcup_R \Gamma^c.$$

Let $\Gamma \sqcup_\epsilon \Gamma^c = (\check{G}, \check{A}, \sigma)$. Then by Definition 24, we have

$$(\check{G}, \check{A}) = (\check{F}, \check{A}) \cup_\epsilon (\check{F}, \check{A})^c,$$

where for all $h \in \check{A}$, we have
$$\check{G}(h) = \check{F}(h) \cup \check{F}^c(h).$$

By Theorem 11, we have

$$\check{G}(h) = \check{F}(h) \cup [\check{F}(h)]^c$$
$$= U(\check{F}(h)).$$

Also, by Theorem 11, we have

$$\sigma = \rho \cup \rho^c$$
$$= U(\rho).$$

Hence,
$$\Gamma \sqcup_\epsilon \Gamma^c = \Gamma \sqcup_R \Gamma^c = \check{U}(\Gamma).$$

□

Similarly, we can obtain the second result.

## 7. A Generalized Picture Fuzzy Soft Sets Based MADM Process

In this section, we defined the expectation score function, Dombi aggregated picture fuzzy decision value ($DAPFDV$), aggregated picture fuzzy decision value ($APFDV$), an algorithm for solving MADM problems and example in support of algorithm.

First, we define the expectation score function of $PFV$, which we use for finding weight vector for PFDWA and PFWA operators. After, we define $DAPFDV$ and $APFDV$, on the basis of which we rank alternatives.

**Definition 32.** *Let* $q = (\xi, \eta, \vartheta)$ *be a PFV. Then the expectation score function* $\check{\delta}$ *is defined as follows:*

$$\check{\delta}(q) = \frac{\xi_q - \vartheta_q + \eta_q + 1}{2}, \ \check{\delta}(q) \in [0,1].$$

**Definition 33.** *Let* $\Gamma = (\check{F}, \check{A}, \rho)$ *be a GPFSS over* $\check{X}$ *such that*

$$\sum_{h \in \check{A}} \check{\delta}_{\tau(h)} = b < +\infty,$$

where $\check{\delta}$ is an expectation score function calculated by Definition 32. Then by using Definition 8, the Dombi aggregated picture fuzzy decision value $(DAPFDV)$ of $f$ in $\check{X}$ is given by

$$W_\Gamma(f) = \oplus_{h \in \check{A}} \frac{\check{\delta}_{\tau(h)}}{b} \check{F}(h)(f),$$

for all $f \in \check{X}$.

**Definition 34.** *Let $\Gamma = (\check{F}, \check{A}, \rho)$ be a GPFSS over $\check{X}$ such that*

$$\sum_{h \in \check{A}} \check{\delta}_{\tau(h)} = b < +\infty,$$

*where $\check{\delta}$ is an expectation score function calculated by Definition 32. Then by Definition 9, the aggregated picture fuzzy decision value $(APFDV)$ of $f$ in $\check{X}$ is given by*

$$Y_\Gamma(f) = \oplus_{h \in \check{A}} \frac{\check{\delta}_{\tau(h)}}{b} \check{F}(h)(f),$$

*for all $f \in \check{X}$.*

The *GPFSS* is used to solve the multi attribute decision making (MADM) problems, where the moderator or director lead the two different groups of experts with their specialties in different fields related to the problem where we make a decision. Experts evaluate the options, choices or alternatives on the basis of criteria of different attributes or characteristics. The following algorithm shows the complete procedure. For simplicity, we assume that all the characteristics are of the beneficial type.

*7.1. Algorithm*

Step 1. Let $\check{X} = \{f_1, f_2, ..., f_n\}$, and $\check{E} = \check{A} \cup \check{B} = \{h_1, h_2, ..., h_n\}$. Two expert groups construct two *BPFSSs* $(\check{F}, \check{A})$ and $(\check{G}, \check{B})$ over $\check{X}$ separately. Two *PPFSs* $\rho$ and $\sigma$ are given by the head or director, which completes the construction of two *GPFSSs* $\Gamma_1 = (\check{F}, \check{A}, \rho)$ and $\Gamma_2 = (\check{G}, \check{B}, \sigma)$.

Step 2. By using Definition 25, calculate extended intersection $\Gamma = \Gamma_1 \sqcap_\epsilon \Gamma_2 = (\check{H}, \check{C}, \tau)$, of $\Gamma_1$ and $\Gamma_2$.

Step 3. Calculate the Dombi aggregated picture fuzzy decision values $(DAPFDVs)$ by using picture fuzzy Dombi weighted average operator $(PFDWA)$ as follows,

$$W_\Gamma(f_i) = \oplus_{j=1}^m \frac{\check{\delta}_{\tau(h_j)}}{b} \check{H}(h_j)(f_i).$$

Step 4. Ascendingly rank $W_\Gamma(f_i)$ according to Definition 7.

Step 5. Rank $f_i$ $(i = 1, 2, 3, ..., n)$ ascendingly according to the rank of $W_\Gamma(f_i)$ and output $f_i$ as the optimal decision if it is the largest *PFV* according to Definition 7.

**Remark 3.** *Section 7.1 is directly applied to the real life problems and we can extend it to the finite number of groups. In this algorithm, we can easily see that the groups which consist of experts (who make proper and effective evaluation on the basis of their experiences) gave BPFSSs, and the head/director (who is the responsible of the firm or department) judge the evaluation made by groups generally and give their opinion in the form of PPFSs, which completes the formulation of GPFSSs. In the third step, we use extended intersection to integrate the information from GPFSSs. Next it is very important that we calculate the weight vector from PPFS by using expectation score function and make proper use of PPFS and after that we calculate DAPFDVs and rank $f_i$ according to the rank of DAPFDVs.*

First, we proceed the calculations for optimal decision in Example 3.

**Example 4.** *Let* $\Gamma_1 = (\check{F}, \check{A}, \rho)$ *and* $\Gamma_2 = (\check{G}, \check{B}, \sigma)$ *be two GPFSS over* $\check{X}$ *which define in Example* 3. *We apply Section* 7.1 *to find an optimal alternative.*

**Step 1.** *First, we find the extended intersection by using Definition* 25.

*Let*
$$\Gamma = (\check{H}, \check{C}, \tau) = (\check{F}, \check{A}, \rho) \sqcap_\epsilon (\check{G}, \check{B}, \sigma).$$

*For all* $h \in \check{C} = \check{A} \cup \check{B}$, *we have*

$$\tau = \{(0.5, 0.1, 0.3)/h_1, (0.3, 0.4, 0.2)/h_2, (0.2, 0.3, 0.4)/h_3, (0.3, 0.2, 0.4)/h_4,$$
$$(0.1, 0.4, 0.4)/h_5\}.$$

*Moreover, we have*

$$\check{H}(h_1) = \{(0.3, 0.2, 0.1)/f_1, (0.4, 0.2, 0.2)/f_2, (0.2, 0.3, 0.4)/f_3, (0.1, 0.2, 0.5)/f_4\},$$

$$\check{H}(h_2) = \{(0.4, 0.5, 0.1)/f_1, (0.2, 0.1, 0.5)/f_2, (0.3, 0.4, 0.2)/f_3, (0.1, 0.6, 0.1)/f_4\},$$

$$\check{H}(h_3) = \{(0.6, 0.1, 0.1)/f_1, (0.2, 0.1, 0.5)/f_2, (0.2, 0.1, 0.4)/f_3, (0.1, 0.5, 0.1)/f_4\},$$

$$\check{H}(h_4) = \{(0.6, 0.2, 0.1)/f_1, (0.5, 0.2, 0.2)/f_2, (0.4, 0.4, 0.2)/f_3, (0.2, 0.4, 0.2)/f_4\},$$

$$\check{H}(h_5) = \{(0.3, 0.2, 0.4)/f_1, (0.6, 0.1, 0.1)/f_2, (0.5, 0.1, 0.3)/f_3, (0.2, 0.2, 0.5)/f_4,\}.$$

*The tabular representation of extended union is shown in Table* 7.

**Table 7.** The *GPFSS* $\Gamma = (\check{H}, \check{A} \cup \check{B}, \tau) = (\check{F}, \check{A}, \rho) \sqcap_\epsilon (\check{G}, \check{B}, \sigma)$.

| $\check{X}$ | $h_1$ | $h_2$ | $h_3$ | $h_4$ | $h_5$ |
|---|---|---|---|---|---|
| $f_1$ | (0.3, 0.2, 0.1) | (0.4, 0.5, 0.1) | (0.6, 0.1, 0.1) | (0.6, 0.2, 0.1) | (0.3, 0.2, 0.4) |
| $f_2$ | (0.4, 0.2, 0.2) | (0.2, 0.1, 0.5) | (0.2, 0.1, 0.5) | (0.5, 0.2, 0.2) | (0.6, 0.1, 0.1) |
| $f_3$ | (0.2, 0.3, 0.4) | (0.3, 0.4, 0.2) | (0.2, 0.1, 0.4) | (0.4, 0.4, 0.2) | (0.5, 0.1, 0.3) |
| $f_4$ | (0.1, 0.2, 0.5) | (0.1, 0.6, 0.1) | (0.1, 0.5, 0.1) | (0.2, 0.4, 0.2) | (0.2, 0.2, 0.5) |
| $\rho$ | (0.5, 0.1, 0.3) | (0.3, 0.4, 0.2) | (0.2, 0.3, 0.4) | (0.3, 0.2, 0.4) | (0.1, 0.4, 0.4) |

**Step 2.** *Now, we calculate Dombi aggregated picture fuzzy decision values* $(DAPFDVs)$ *by Definition* 33, *using PFDWA for* $k = 1$. *First, we calculate weight vectors from the picture fuzzy set by using expectation score function* $\check{\delta}_{\tau(h_j)}$ $(j = 1, 2, ..., 5)$ *using Definition* 32, *where the expectation score functions are* $\check{\delta}_1 = 0.65$, $\check{\delta}_2 = 0.75$, $\check{\delta}_3 = 0.55$, $\check{\delta}_4 = 0.55$, $\check{\delta}_5 = 0.55$ *and their sum is* $b = \sum_{h \in \check{A}} \check{\delta}_{\tau(h)} = 3.05$. *Following is the weight vector*

$$\check{\omega} = (0.2131, 0.2459, 0.1803, 0.1803, 0.1803)^T,$$

*which is calculated from the formula* $\check{\omega}_j = \frac{\check{\delta}_{\tau(h_j)}}{b}$ $(j = 1, 2, ..., 5)$, *where* $b = \sum_{h \in \check{A}} \check{\delta}_{\tau(h)}$. *More detail founds in Table* 8.

**Table 8.** Weights calculated from the *PFS* $\tau$.

| $\check{X}$ | $h_1$ | $h_2$ | $h_3$ | $h_4$ | $h_5$ |
|---|---|---|---|---|---|
| $\tau$ | (0.5, 0.1, 0.3) | (0.3, 0.4, 0.2) | (0.2, 0.3, 0.4) | (0.3, 0.2, 0.4) | (0.1, 0.4, 0.4) |
| $\check{\delta}_{\tau(h_j)}$ | 0.65 | 0.75 | 0.55 | 0.55 | 0.55 |
| $\check{\omega}_j$ | 0.2131 | 0.2459 | 0.1803 | 0.1803 | 0.1803 |

*Now using these weight vector, the DAPFDVs can be calculated as:*

$$W_\Gamma(f_i) = PFDWA_{\check{\omega}}(\check{H}(h_1)(f_i), \check{H}(h_2)(f_i), \check{H}(h_3)(f_i), \check{H}(h_4)(f_i), \check{H}(h_5)(f_i))$$

$$= \oplus_{j=1}^{5} \check{\omega}_j \check{H}(h_j)(f_i).$$

*So, the DAPFDVs are*

$$W_\Gamma(f_1) = (0.46622, 0.19367, 0.11565),$$

$$W_\Gamma(f_2) = (0.41155, 0.12450, 0.21633),$$

$$W_\Gamma(f_3) = (0.33521, 0.18581, 0.26914),$$

$$W_\Gamma(f_4) = (0.13881, 0.31365, 0.16806).$$

**Step 3.** *Find score function of $W_\Gamma(f_i)$ $(i = 1, 2, 3, 4)$ as*

$$\check{\Theta}(W_\Gamma(f_1)) = 0.466221 - 0.115649 = 0.350572.$$

*Similarly, we get $\check{\Theta}(W_\Gamma(f_2)) = 0.195212$, $\check{\Theta}(W_\Gamma(f_3)) = 0.066075$ and $\check{\Theta}(W_\Gamma(f_4)) = -0.029246$. More detail founds in Table 9.*

**Table 9.** Dombi aggregated picture fuzzy decision values ($DAPFDV$) and score functions.

| $\check{X}$ | $DAPFDVs$ | $\check{\Theta}(W_\Gamma(f_i))$ |
|---|---|---|
| $f_1$ | (0.46622, 0.19367, 0.11564) | 0.35057 |
| $f_2$ | (0.41155, 0.12450, 0.21633) | 0.19521 |
| $f_3$ | (0.33521, 0.18581, 0.26914) | 0.06607 |
| $f_4$ | (0.13881, 0.31365, 0.16806) | −0.02925 |

**Step 4.** *Ranking the DAPFDVs according to Definition 7, we have*

$$\check{\Theta}(W_\Gamma(f_4)) \prec \check{\Theta}(W_\Gamma(f_3)) \prec \check{\Theta}(W_\Gamma(f_2)) \prec \check{\Theta}(W_\Gamma(f_1)).$$

**Step 5.** *From above calculations, alternatives have the order*

$$f_4 \prec f_3 \prec f_2 \prec f_1.$$

*Hence $f_1$ is the most suitable/optimal choice for the customer.*

**Remark 4.** *For consistency, when we use the $k = 2$ and $k = 3$, still we have the $f_1$ optimal. Details are in Tables 10 and 11.*

**Table 10.** DAPFDVs and score functions for $k = 2$.

| $\check{X}$ | $DAPFDVs$ | $\check{\Theta}(W_\Gamma(f_i))$ |
|---|---|---|
| $f_1$ | (0.49911, 0.16944, 0.10902) | 0.39009 |
| $f_2$ | (0.45682, 0.11842, 0.17800) | 0.27882 |
| $f_3$ | (0.36493, 0.15158, 0.25334) | 0.11158 |
| $f_4$ | (0.14853, 0.27444, 0.13990) | 0.00864 |

**Table 11.** DAPFDVs and score functions for $k = 3$.

| $\check{X}$ | *DAPFDVs* | $\check{\Theta}(W_\Gamma(f_i))$ |
|---|---|---|
| $f_1$ | (0.52317, 0.15332, 0.10610) | 0.41708 |
| $f_2$ | (0.48730, 0.11415, 0.15637) | 0.33093 |
| $f_3$ | (0.38935, 0.13442, 0.24167) | 0.14768 |
| $f_4$ | (0.15735, 0.25227, 0.12725) | 0.03010 |

**Remark 5.** *In Algorithm 7.1, if we calculate the aggregated picture fuzzy decision values* (*APFDVs*) *by using picture fuzzy weighted averaging operator* (*PFWA*) *as follows,*

$$Y_\Gamma(f_i) = \oplus_{j=1}^m \frac{\check{\delta}_{\tau(h_j)}}{b} \check{H}(h_j)(f_i).$$

*Next, rank ascendingly $Y_\Gamma(f_i)$ according to the Definition 7, then rank $f_i$ ($i = 1,2,3,4$) ascendingly and output $f_i$ as the optimal decision if it is the largest PFV according to Definition 7.*

**Example 5.** *In Example 4, we calculate APFDVs according to Definition 34, then again we get the $f_1$ is optimal or best choice. Detail founds in Table 12.*

$$\check{\Theta}(Y_\Gamma(f_4)) \prec \check{\Theta}(Y_\Gamma(f_3)) \prec \check{\Theta}(Y_\Gamma(f_2)) \prec \check{\Theta}(Y_\Gamma(f_1)).$$

$$f_4 \prec f_3 \prec f_2 \prec f_1.$$

**Table 12.** *APFDVs* and score functions.

| $\check{X}$ | *APFDVs* | $\check{\Theta}(Y_\Gamma(f_i))$ |
|---|---|---|
| $f_1$ | (0.44918, 0.22115, 0.12843) | 0.32075 |
| $f_2$ | (0.38991, 0.13138, 0.26087) | 0.12904 |
| $f_3$ | (0.32468, 0.22823, 0.28267) | 0.04201 |
| $f_4$ | (0.13742, 0.35031, 0.21348) | −0.07606 |

**Remark 6.** *Since two types of criteria occur in GPFSS* $(\check{F}, \check{A}, \rho)$, *namely, cost and benefit criteria. So, for consolidation, we must normalize the* $(\check{F}, \check{A}, \rho)$ *through the following equation:*

$$q = \begin{cases} (\xi_q, \eta_q, \vartheta_q), & \text{if } q \text{ is a benifit criteria,} \\ (\vartheta_q, \eta_q, \xi_q), & \text{if } q \text{ is a cost criteria,} \end{cases}$$

*such that the normalized GPFSS is denoted by* $(\check{F}', \check{A}, \rho')$, *where* $(\check{F}', \check{A})$ *is the normalization of BPFSS* $(\check{F}, \check{A})$ *and $\rho'$ is the normalization of PPFS $\rho$.*

## 8. Case Study: A Tower Construction Problem

A private bank wants to build a tower of height 400 m. It involves a very complicated evaluation and decision-making because it is a very big project. The construction company may be examined by different attributes like "credentials", "modern equipment and technology" and so forth. To chose the felicitous alternative the director to consult with experts for their professional opinions.

Suppose $\check{X} = \{f_1, f_2, f_3, f_4, f_5, f_6, f_7, f_8\}$, be the top eight world construction companies. For felicitous choice the director who is head of the committee which contains the experts from different departments like architecture, engineering, management, construction, finance management and planing departments. The committee evaluated the company on the basis of the following criteria $\check{E} = \{h_1, h_2, h_3, h_4, h_5, h_6\}$, where $h_j$ stands for "credentials", "modern equipment and technology", "a skilled team", "cost", "strong risk management" and "rich portfolios", respectively. The director

divides the committee into two groups to do the evaluation. The set of attributes $\check{A} = \{h_2, h_3, h_4, h_6\}$ is assigned to the first group and $\check{B} = \{h_1, h_2, h_3, h_5\}$ is given to the second group. These two groups evaluate the alternatives (companies) and gives the *PFSSs* $(\check{F}, \check{A})$ and $(\check{G}, \check{B})$ accordingly. The director scrutinizes the work done by two expert groups generally and gives the two *PFSs* $\rho$ and $\sigma$ that complete the constructions of two *GPFSSs* $(\check{F}, \check{A}, \rho)$ and $(\check{G}, \check{B}, \sigma)$ as shown in Tables 13 and 14.

**Table 13.** The *GPFSS* $(\check{F}, \check{A}, \rho)$.

| $\check{X}$ | $h_2$ | $h_3$ | $h_4$ | $h_6$ |
|---|---|---|---|---|
| $f_1$ | (0.4, 0.3, 0.2) | (0.2, 0.3, 0.4) | (0.5, 0.1, 0.3) | (0.3, 0.2, 0.5) |
| $f_2$ | (0.3, 0.4, 0.3) | (0.3, 0.2, 0.4) | (0.4, 0.2, 0.3) | (0.4, 0.1, 0.4) |
| $f_3$ | (0.2, 0.2, 0.5) | (0.5, 0.3, 0.1) | (0.6, 0.1, 0.2) | (0.4, 0.2, 0.3) |
| $f_4$ | (0.5, 0.1, 0.3) | (0.6, 0.1, 0.2) | (0.1, 0.1, 0.7) | (0.2, 0.4, 0.3) |
| $f_5$ | (0.6, 0.1, 0.2) | (0.2, 0.2, 0.5) | (0.2, 0.2, 0.5) | (0.2, 0.2, 0.4) |
| $f_6$ | (0.2, 0.2, 0.5) | (0.1, 0.3, 0.4) | (0.3, 0.1, 0.4) | (0.5, 0.1, 0.3) |
| $f_7$ | (0.3, 0.1, 0.5) | (0.3, 0.3, 0.3) | (0.3, 0.2, 0.4) | (0.3, 0.1, 0.5) |
| $f_8$ | (0.4, 0.2, 0.3) | (0.4, 0.2, 0.3) | (0.5, 0.3, 0.1) | (0.4, 0.1, 0.5) |
| $\rho$ | (0.4, 0.2, 0.3) | (0.5, 0.1, 0.3) | (0.3, 0.2, 0.5) | (0.2, 0.2, 0.5) |

**Table 14.** The *GPFSS* $(\check{G}, \check{B}, \sigma)$.

| $\check{X}$ | $h_1$ | $h_2$ | $h_3$ | $h_5$ |
|---|---|---|---|---|
| $f_1$ | (0.1, 0.3, 0.5) | (0.2, 0.3, 0.4) | (0.3, 0.2, 0.5) | (0.4, 0.1, 0.4) |
| $f_2$ | (0.5, 0.1, 0.3) | (0.5, 0.1, 0.3) | (0.6, 0.1, 0.2) | (0.3, 0.2, 0.5) |
| $f_3$ | (0.2, 0.4, 0.3) | (0.3, 0.3, 0.3) | (0.5, 0.3, 0.1) | (0.5, 0.1, 0.3) |
| $f_4$ | (0.6, 0.1, 0.2) | (0.1, 0.3, 0.4) | (0.2, 0.2, 0.5) | (0.2, 0.2, 0.5) |
| $f_5$ | (0.2, 0.2, 0.5) | (0.3, 0.2, 0.4) | (0.4, 0.1, 0.3) | (0.6, 0.1, 0.2) |
| $f_6$ | (0.5, 0.1, 0.3) | (0.6, 0.2, 0.1) | (0.3, 0.2, 0.3) | (0.3, 0.1, 0.5) |
| $f_7$ | (0.4, 0.2, 0.3) | (0.4, 0.3, 0.2) | (0.7, 0.1, 0.2) | (0.4, 0.2, 0.3) |
| $f_8$ | (0.3, 0.4, 0.3) | (0.1, 0.5, 0.2) | (0.2, 0.3, 0.4) | (0.3, 0.1, 0.4) |
| $\sigma$ | (0.4, 0.2, 0.3) | (0.3, 0.4, 0.3) | (0.4, 0.1, 0.4) | (0.5, 0.2, 0.3) |

Since the attribute $h_4$ involves the cost criteria, therefore, we have to normalize the $(\check{F}, \check{A}, \rho)$ using Remark 6, as shown in Table 15.

**Table 15.** The *GPFSS* $(\check{F}', \check{A}, \rho')$.

| $\check{X}$ | $h_2$ | $h_3$ | $h_4$ | $h_6$ |
|---|---|---|---|---|
| $f_1$ | (0.4, 0.3, 0.2) | (0.2, 0.3, 0.4) | (0.3, 0.1, 0.5) | (0.3, 0.2, 0.5) |
| $f_2$ | (0.3, 0.4, 0.3) | (0.3, 0.2, 0.4) | (0.3, 0.2, 0.4) | (0.4, 0.1, 0.4) |
| $f_3$ | (0.2, 0.2, 0.5) | (0.5, 0.3, 0.1) | (0.2, 0.1, 0.6) | (0.4, 0.2, 0.3) |
| $f_4$ | (0.5, 0.1, 0.3) | (0.6, 0.1, 0.2) | (0.7, 0.1, 0.1) | (0.2, 0.4, 0.3) |
| $f_5$ | (0.6, 0.1, 0.2) | (0.2, 0.2, 0.5) | (0.5, 0.2, 0.2) | (0.2, 0.2, 0.4) |
| $f_6$ | (0.2, 0.2, 0.5) | (0.1, 0.3, 0.4) | (0.4, 0.1, 0.3) | (0.5, 0.1, 0.3) |
| $f_7$ | (0.3, 0.1, 0.5) | (0.3, 0.3, 0.3) | (0.4, 0.2, 0.3) | (0.3, 0.1, 0.5) |
| $f_8$ | (0.4, 0.2, 0.3) | (0.4, 0.2, 0.3) | (0.1, 0.3, 0.5) | (0.4, 0.1, 0.5) |
| $\rho'$ | (0.4, 0.2, 0.3) | (0.5, 0.1, 0.3) | (0.5, 0.2, 0.3) | (0.2, 0.2, 0.5) |

Now, in the second step we integrate the above information by using extended intersection according to Definition 25, as

$$\Gamma = (\check{F}', \check{A}, \rho') \sqcap_\epsilon (\check{G}, \check{B}, \sigma),$$

detail founds in Table 16.

**Table 16.** The *GPFSS* $\Gamma = (\check{F}', \check{A}, \rho') \sqcap_\epsilon (\check{G}, \check{B}, \sigma)$.

| $\check{X}$ | $h_1$ | $h_2$ | $h_3$ | $h_4$ | $h_5$ | $h_6$ |
|---|---|---|---|---|---|---|
| $f_1$ | (0.1, 0.3, 0.5) | (0.2, 0.3, 0.4) | (0.2, 0.2, 0.5) | (0.3, 0.1, 0.5) | (0.4, 0.1, 0.4) | (0.3, 0.2, 0.5) |
| $f_2$ | (0.5, 0.1, 0.3) | (0.3, 0.1, 0.3) | (0.3, 0.1, 0.4) | (0.3, 0.2, 0.4) | (0.3, 0.2, 0.5) | (0.4, 0.1, 0.4) |
| $f_3$ | (0.2, 0.4, 0.3) | (0.2, 0.2, 0.5) | (0.5, 0.3, 0.1) | (0.2, 0.1, 0.6) | (0.5, 0.1, 0.3) | (0.4, 0.2, 0.3) |
| $f_4$ | (0.6, 0.1, 0.2) | (0.1, 0.1, 0.4) | (0.2, 0.1, 0.5) | (0.7, 0.1, 0.1) | (0.2, 0.2, 0.5) | (0.2, 0.4, 0.3) |
| $f_5$ | (0.2, 0.2, 0.5) | (0.3, 0.1, 0.4) | (0.2, 0.1, 0.5) | (0.5, 0.2, 0.2) | (0.6, 0.1, 0.2) | (0.2, 0.2, 0.4) |
| $f_6$ | (0.5, 0.1, 0.3) | (0.2, 0.2, 0.5) | (0.1, 0.2, 0.4) | (0.4, 0.1, 0.3) | (0.3, 0.1, 0.5) | (0.5, 0.1, 0.3) |
| $f_7$ | (0.4, 0.2, 0.3) | (0.3, 0.1, 0.5) | (0.3, 0.1, 0.3) | (0.4, 0.2, 0.3) | (0.4, 0.2, 0.3) | (0.3, 0.1, 0.5) |
| $f_8$ | (0.3, 0.4, 0.3) | (0.1, 0.2, 0.3) | (0.2, 0.2, 0.4) | (0.1, 0.3, 0.5) | (0.3, 0.1, 0.4) | (0.4, 0.1, 0.5) |
| $\tau$ | (0.4, 0.2, 0.3) | (0.3, 0.2, 0.3) | (0.4, 0.1, 0.4) | (0.5, 0.2, 0.3) | (0.5, 0.2, 0.3) | (0.2, 0.2, 0.5) |

From the *PFS* $\tau$, we calculate weight vector from picture fuzzy sets by using expectation score function $\check{\delta}_{\tau(h_j)}$ ($j = 1, 2, ..., 6$) using Definition 32, where the expectation score functions are $\check{\delta}_1 = 0.65$, $\check{\delta}_2 = 0.6$, $\check{\delta}_3 = 0.55$, $\check{\delta}_4 = 0.7$, $\check{\delta}_5 = 0.7$, $\check{\delta}_6 = 0.45$ and their sum $b = \sum_{h \in \check{A}} \check{\delta}_{\tau(h)} = 3.65$. Following weight vector are

$$\check{\omega} = (0.1780, 0.1644, 0.1507, 0.1918, 0.1918, 0.1233)^T,$$

which is calculated from the formula $\check{\omega}_j = \frac{\check{\delta}_{\tau(h_j)}}{b}$ ($j = 1, 2, ..., 6$), where $b = \sum_{h \in \check{A}} \check{\delta}_{\tau(h)}$. More details are found in Table 17.

**Table 17.** Weights calculated from the *PFS* $\tau$.

| $\check{X}$ | $h_1$ | $h_2$ | $h_3$ | $h_4$ | $h_5$ | $h_6$ |
|---|---|---|---|---|---|---|
| $\tau$ | (0.4, 0.2, 0.3) | (0.3, 0.2, 0.3) | (0.4, 0.1, 0.4) | (0.5, 0.2, 0.3) | (0.5, 0.2, 0.3) | (0.2, 0.2, 0.5) |
| $\check{\delta}_{\tau(h_j)}$ | 0.65 | 0.6 | 0.55 | 0.7 | 0.7 | 0.45 |
| $\check{\omega}_j$ | 0.1780 | 0.1644 | 0.1507 | 0.1918 | 0.1918 | 0.1233 |

Now, using this weight vector, we calculate Dombi aggregated picture fuzzy decision values (*DAPFDVs*) using Definition 33, for $k = 1$. The *DAPFDVs* can be calculated as:

$$W_\Gamma(f_i) = PFDWA_{\check{\omega}}(\check{H}(h_1)(f_i), \check{H}(h_2)(f_i), \check{H}(h_3)(f_i), \check{H}(h_4)(f_i), \check{H}(h_5)(f_i), \check{H}(h_6)(f_i))$$

$$= \oplus_{j=1}^6 \check{\omega}_j \check{H}(h_j)(f_i).$$

So, the *DAPFDVs* are

$$W_\Gamma(f_1) = (0.265496, 0.157546, 0.459116),$$

$$W_\Gamma(f_2) = (0.358828, 0.123732, 0.371826),$$

$$W_\Gamma(f_3) = (0.358254, 0.160724, 0.263218),$$

$$W_\Gamma(f_4) = (0.459240, 0.123209, 0.231749),$$

$$W_\Gamma(f_5) = (0.398662, 0.132723, 0.303522),$$

$$W_\Gamma(f_6) = (0.362737, 0.118701, 0.365923),$$

$$W_\Gamma(f_7) = (0.359912, 0.139043, 0.339014),$$

$$W_\Gamma(f_8) = (0.241239, 0.172092, 0.380549).$$

We find score function of $W_\Gamma(f_i)$ ($i = 1, 2, ..., 8$) as

$$\check{\Theta}(W_\Gamma(f_1)) = 0.265496 - 0.459116 = -0.19362.$$

Similarly, we get $\check{\Theta}(W_\Gamma(f_2)) = -0.012998$, $\check{\Theta}(W_\Gamma(f_3)) = 0.0950365$, $\check{\Theta}(W_\Gamma(f_4)) = 0.227491$, $\check{\Theta}(W_\Gamma(f_5)) = 0.0951392$, $\check{\Theta}(W_\Gamma(f_6)) = -0.00318571$, $\check{\Theta}(W_\Gamma(f_7)) = 0.0208985$ and $\check{\Theta}(W_\Gamma(f_8)) = -0.13931$. Details are found in Table 18.

**Table 18.** DAPFDVs and score functions.

| $\check{X}$ | *DAPFDVs* | $\check{\Theta}(W_\Gamma(f_i))$ |
|---|---|---|
| $f_1$ | (0.265496, 0.157546, 0.459116) | $-0.193620$ |
| $f_2$ | (0.358828, 0.123732, 0.371826) | $-0.012998$ |
| $f_3$ | (0.358254, 0.160724, 0.263218) | 0.095037 |
| $f_4$ | (0.459240, 0.123209, 0.231749) | 0.227491 |
| $f_5$ | (0.398662, 0.132723, 0.303522) | 0.095139 |
| $f_6$ | (0.362737, 0.118701, 0.365923) | $-0.003186$ |
| $f_7$ | (0.359912, 0.139043, 0.339014) | 0.020899 |
| $f_8$ | (0.241239, 0.172092, 0.380549) | $-0.139310$ |

We rank *DAPFDVs* ascendingly using Definition 7, we have

$$\check{\Theta}(W_\Gamma(f_1)) \prec \check{\Theta}(W_\Gamma(f_8)) \prec \check{\Theta}(W_\Gamma(f_2)) \prec \check{\Theta}(W_\Gamma(f_6)) \prec \check{\Theta}(W_\Gamma(f_7)) \prec \check{\Theta}(W_\Gamma(f_3)) \prec$$
$$\check{\Theta}(W_\Gamma(f_5)) \prec \check{\Theta}(W_\Gamma(f_4)).$$

From above calculations, alternatives have the order

$$f_1 \prec f_8 \prec f_2 \prec f_6 \prec f_7 \prec f_3 \prec f_5 \prec f_4.$$

Hence $f_4$ is the most suitable/optimal alternative (construction company) for the bank.

**Remark 7.** *For consistency, when we use the different values for $k = 1, 2, 3, ..., 10$, still we have the $f_4$ optimal alternative. We can see from Table 19, when we change the values of parameter $k$, the order of alternatives respond but for $k \geq 4$, the order becomes smooth and $f_4$ remains optimal for all values of the parameter $k \geq 1$. Details found in Tables 19 and 20.*

**Table 19.** Rank of alternatives.

| $k \geq 1$ | Rank |
|---|---|
| $k = 1$ | $f_1 \prec f_8 \prec f_2 \prec f_6 \prec f_7 \prec f_3 \prec f_5 \prec f_4$ |
| $k = 2$ | $f_1 \prec f_8 \prec f_2 \prec f_7 \prec f_6 \prec f_5 \prec f_3 \prec f_4$ |
| $k = 3$ | $f_1 \prec f_8 \prec f_2 \prec f_7 \prec f_6 \prec f_5 \prec f_3 \prec f_4$ |
| $k = 4$ | $f_1 \prec f_8 \prec f_7 \prec f_2 \prec f_6 \prec f_5 \prec f_3 \prec f_4$ |
| $k = 5$ | $f_1 \prec f_8 \prec f_7 \prec f_2 \prec f_6 \prec f_5 \prec f_3 \prec f_4$ |
| $k = 6$ | $f_1 \prec f_8 \prec f_7 \prec f_2 \prec f_6 \prec f_5 \prec f_3 \prec f_4$ |
| $k = 7$ | $f_1 \prec f_8 \prec f_7 \prec f_2 \prec f_6 \prec f_5 \prec f_3 \prec f_4$ |
| $k = 8$ | $f_1 \prec f_8 \prec f_7 \prec f_2 \prec f_6 \prec f_5 \prec f_3 \prec f_4$ |
| $k = 9$ | $f_1 \prec f_8 \prec f_7 \prec f_2 \prec f_6 \prec f_5 \prec f_3 \prec f_4$ |
| $k = 10$ | $f_1 \prec f_8 \prec f_7 \prec f_2 \prec f_6 \prec f_5 \prec f_3 \prec f_4$ |

**Table 20.** Score functions of DAPFDVs.

| $k \geq 1$ | $f_1$ | $f_2$ | $f_3$ | $f_4$ | $f_5$ | $f_6$ | $f_7$ | $f_8$ |
|---|---|---|---|---|---|---|---|---|
| $k = 1$ | $-0.19362$ | $-0.01299$ | $0.09503$ | $0.22749$ | $0.09513$ | $-0.00318$ | $0.02089$ | $-0.13931$ |
| $k = 2$ | $-0.16522$ | $0.01328$ | $0.19178$ | $0.36488$ | $0.17957$ | $0.04482$ | $0.03606$ | $-0.09816$ |
| $k = 3$ | $-0.14220$ | $0.03890$ | $0.25160$ | $0.43441$ | $0.23462$ | $0.07727$ | $0.04737$ | $-0.06694$ |
| $k = 4$ | $-0.12349$ | $0.06179$ | $0.28713$ | $0.47314$ | $0.26955$ | $0.09985$ | $0.05584$ | $-0.04302$ |
| $k = 5$ | $-0.11259$ | $0.07497$ | $0.29812$ | $0.48843$ | $0.28479$ | $0.11045$ | $0.05913$ | $-0.03117$ |
| $k = 6$ | $-0.09550$ | $0.09665$ | $0.32461$ | $0.51417$ | $0.30931$ | $0.12821$ | $0.06734$ | $-0.00906$ |
| $k = 7$ | $-0.08501$ | $0.10925$ | $0.33545$ | $0.52632$ | $0.32153$ | $0.13747$ | $0.07132$ | $0.00328$ |
| $k = 8$ | $-0.07625$ | $0.11945$ | $0.34360$ | $0.53555$ | $0.33093$ | $0.14474$ | $0.07451$ | $0.01346$ |
| $k = 9$ | $-0.06890$ | $0.12778$ | $0.34995$ | $0.54278$ | $0.33839$ | $0.15058$ | $0.07712$ | $0.02194$ |
| $k = 10$ | $-0.06270$ | $0.13465$ | $0.35502$ | $0.54859$ | $0.34444$ | $0.15535$ | $0.07928$ | $0.02906$ |

**Example 6.** *In case Section 8, we calculate APFDVs according to Definition 34, using PFWA, then again we get the $f_4$ is an optimal or best choice for construction, and the order of the alternatives is*

$$f_1 \prec f_8 \prec f_6 \prec f_2 \prec f_7 \prec f_3 \prec f_5 \prec f_4.$$

*Detail founds in Table 21.*

**Table 21.** *APFDVs* and score functions.

| $\check{X}$ | *DAPFDVs* | $\breve{\Theta}(W_{\Gamma}(f_i))$ |
|---|---|---|
| $f_1$ | $(0.258760, 0.176136, 0.461797)$ | $-0.203036$ |
| $f_2$ | $(0.353107, 0.130459, 0.378328)$ | $-0.025221$ |
| $f_3$ | $(0.342684, 0.184364, 0.315812)$ | $0.026872$ |
| $f_4$ | $(0.402669, 0.135510, 0.282350)$ | $0.120318$ |
| $f_5$ | $(0.373864, 0.140747, 0.329943)$ | $0.043921$ |
| $f_6$ | $(0.348055, 0.124409, 0.375813)$ | $-0.027759$ |
| $f_7$ | $(0.358051, 0.147591, 0.347494)$ | $0.010557$ |
| $f_8$ | $(0.233592, 0.196577, 0.388882)$ | $-0.155289$ |

## 9. Comparison

1. First, we compare our method with the method proposed in [40]. In his paper he did not give any information about how he calculated the weight vector, but in our proposed method we give a proper way to find the weight vector by using the expectation score function $\check{\omega} = \check{\delta}_{\tau(h)} / \sum_{h \in \check{A}} \check{\delta}_{\tau(h)}$. For this, we actually use the parametric picture fuzzy soft sets (*PPFSs*), $\rho$ and $\sigma$ which are given by the head or director who is responsible for firm or department in the form of *PFSs*, which is actually an additional judgment about the general quality of work done by the specialists groups.

2. Secondly, if we compare our method with the method proposed in [30], we also find that they did not give any information about the weight vector. Also, in case Section 8, when we use the operator defined in [30], we get the same optimal decision and in addition, we are working in a more general situation.

3. In [32], De Morgan's laws hold with restricted conditions, while in this paper we relaxed the conditions for De Morgan's laws by defining the new operations, like the extended union, extended intersection, restricted union, and restricted intersection.

4. Our proposed algorithm is related to the picture fuzzy environment while the methods proposed in [17,18,41,42] deal with the intuitionistic fuzzy environment, generalized intuitionistic fuzzy soft environment, and single-valued neutrosophic environment but not in picture fuzzy environment.

## 10. Conclusions

In this paper, we investigate the basic properties of picture fuzzy soft sets, defined more generalized operations of picture fuzzy soft sets and relaxed the conditions for De Morgan's laws for these operations. We proposed a generalized picture fuzzy soft set by combining the picture fuzzy soft set and picture fuzzy set. We introduced some basic notions of generalized picture fuzzy soft sets and defined some operations of generalized picture fuzzy soft sets and also proved De Morgan's laws for these operations. We define upper and lower substitutions for generalized picture fuzzy soft set and prove some important results related to upper and lower substitutions. We proposed an algorithm for solving MADM problems by using extended intersection for generalized picture fuzzy soft information and picture fuzzy Dombi weighted average ($PFDWA$) operator, where we introduced a proper method to obtain the weight vector by using the expectation score function. Then we gave an example and case study of building a tower, where we used the proposed algorithm and got the optimal alternative. Also, we use the picture fuzzy weighted averaging $PFWA$ operator for both example and case study and reached the same results. For consistency, we used different values of $k \geq 1$ in $PFDWA$ and found the same optimal alternative. We have compared our proposed algorithm with previously proposed methods and found it to be more generalized and effective over all the existing structures of fuzzy soft sets. In future work, our proposed set and algorithm can be used to solve MADM problems, risk evaluation, and some other situations under uncertainty environments. For future work, it will be interesting to develop some new techniques, to deal with multi-attribute classification, such as personal evaluation, medical artificial intelligence, energy management and supplier selection evaluation using generalized picture fuzzy soft sets.

**Author Contributions:** The authors contributed equally to writing this article. All authors read and approved the final manuscript.

**Funding:** Petchra Pra Jom Klao Doctoral Scholarship for Ph.D. program of King Mongkut's University of Technology Thonburi (KMUTT) and Theoretical and Computational Science (TaCS) Center. The Rajamangala University of Technology Thanyaburi (RMUTT) (Grant No. NSF62D0604).

**Acknowledgments:** This project was supported by Theoretical and Computational Science (TaCS) Center under Computational and Applied Science for Smart Innovation research Cluster (CLASSIC), Faculty of Science, KMUTT. The first author gives thanks for the support of the Petchra Pra Jom Klao Doctoral Scholarship Academic for Ph.D. Program at King Mongkut's University of Technology Thonburi (KMUTT). Moreover, this research work was financially supported by King Mongkut's University of Technology Thonburi through the KMUTT 55th Anniversary Commemorative Fund. The second author was supported by the Thailand Research Fund (TRF) and the King Mongkut's University of Technology Thonburi (KMUTT) under TRF Research Scholar Award (Grant No. RSA6080047). Furthermore, Wiyada Kumam was financially supported by the Rajamangala University of Technology Thanyaburi (RMUTT) (Grant No. NSF62D0604).

**Conflicts of Interest:** The authors declare no conflict of interest.

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
