# Peer review of "Generalized Picture Fuzzy Soft Sets and Their Application in Decision Support Systems"

_symmetry, doi:10.3390/sym11030415_

Round 1
Reviewer 1 Report
The paper discusses the use of fuzzy soft sets in decision support applications. The introduction provides a sufficient and up to date literature review. I think that sections 2 and 3 are well presented. I have an observation related to the properties of the "picture fuzzy soft sets" - can you please present them in more detail and underline the need of using this type of fuzzy sets. Please present the limitations of your study and how it can be extended in the future (please add this part in the conclusion section).
Please remove the rectangles which are very much presented within the text (e.g. page 6, row 139). Remove ";" in page 7, row 164. Thank you!
Author Response
Dear Reviewer,
Point-by-point Response to the reviewers comments
Reviewer 1
Reviewer: The paper discusses the use of fuzzy soft sets in decision support
applications. The introduction provides a sufficient and up to date literature
review. I think that sections 2 and 3 are well presented.
Authors: We thank the reviewer for this appreciation and their thoughtful
comments and recommendations. We have carefully addressed the reviewers
suggestions and in doing so feel the manuscript is substantially strengthened.
Reviewer: I have an observation related to the properties of the ”picture
fuzzy soft sets” can you please present them in more detail and underline the
need of using this type of fuzzy sets.
Authors: I have given more details of PFSSs in page 5 (lines 131-134, 138-
147) of the revised manuscript.
Reviewer: Please present the limitations of your study and how it can be extended in the future (please add this part in the conclusion section).
Authors: I have done according to given suggestions and mention some direction that will be fruitful using proposed technique to deal with uncertainty in different areas of decision theory.
Reviewer: Please remove the rectangles which are very much presented within the text (e.g. page 6, row 139). Remove “:” in page 7, row 164.
Authors: I have removed all the rectangles and “;” where necessary in revised manuscript.
Best regards,
Poom Kumam

Reviewer 2 Report
General comments for improvement:
1. The language requires very extensive editing.
2. We need more descriptions along the text, for example, section 2 is an array of definitions and the reader should be informed of why they are introduced and for what use they are.
3. The references are not correctly ordered.
Major comment.
I am suspicious about the main definition, which is Definition 17. The so-called the parametric PFS is said to be a PFS in A. However it looks like a PFSS (Def. 10), because it maps the element in A to elements in PF(A). Actually, Example 2 describes the parametric PFS as a PFS. So, Definition 17 must be corrected.
The first paragraph of Section 4 is intended to motivate the need for this Definition 17. It seems very naive. Please clarify what the new concept is meant to express.
Minor comments.
The paragraph starting at line 24 describes hybrid models with soft sets and other concepts. Line 26 presents intuitionistic fuzzy sets, which is not one of these hybrid models. They should be presented before line 24. In addition, there are other remarkable models that should be cited for completeness:
Hesitant Fuzzy Soft Set and Its Applications in Multicriteria Decision Making. Journal of Applied Mathematics Volume 2014, Article ID 643785
Example 2: the parametric PFS describes the view of an expert on what?
Typos and minor issues.
Line 26: delete one of the two “the the”
Line 150: scrutinizes.
Line 157: “we call …. soft set”.
Definition 5: Then their containment ….
Definition 6: we should know what a picture fuzzy value is before using it. This could be done in Definition 4 easily.
Definition 10: we should know what PF(X) is. Surely it means the set of all picture fuzzy sets on X, but I think that this is not defined anywhere.
Author Response
Dear Riviewer,
Reviewer 2
General comments for improvement:
Reviewer: The language requires very extensive editing.
Authors: I have extensively edited the language by using some on-line gram-
mar checkers such as “Grammarly”, “Language tool proofreading software”,
etc,. In addition, some colleagues that are native English speakers also went
through and make some corrections to improve the language.
Reviewer: We need more descriptions along the text, for example, section
2 is an array of definitions and the reader should be informed of why they
are introduced and for what use they are.
Authors: We have given more description of every definition as suggested
by the reviewer. This can found in page 3 (lines 76-79, 82-86, 89-93, 96-100,
107), page 4 (lines 114, 122-124, 126-128), page 5 (lines 131-134, 138-142,
149-150), page 6 (lines 167-168), page 8 (lines 196-205),page 9 (lines 209-
215, 219-220), page 10 (lines 238-241) and page 16 (lines 324-329) of the
revised manuscript.
Reviewer: The references are not correctly ordered.
Authors: We have given order to the references in the revised manuscript.
Major comment:
Reviewer: I am suspicious about the main definition, which is Definition
17. The so called the parametric PFS is said to be a PFS in A. However it
looks like a PFSS (Def. 10), because it maps the element in A to elements
in PF(A). Actually, Example 2 describes the parametric PFS as a PFS.
So, Definition 17 must be corrected.
Authors: I have given more description in page 9 (lines 209-215), which
shows the major difference between (Def. 10) and (Def. 19) in the revised
manuscript.
Reviewer: The first paragraph of Section 4 is intended to motivate the need
Ror this Definition 17. It seems very naive. Please clarify what the new con-
cept is meant to express.
Authors: I rewrite the first paragraph of Section 4 on page 8 (lines 196-205)
to include the motivation of Definition 19 in the revised manuscript.
Minor comments:
Reviewer: The paragraph starting at line 24 describes hybrid models with
soft sets and other concepts. Line 26 presents intuitionistic fuzzy sets, which
is not one of these hybrid models. They should be presented before line
24. In addition, there are other remarkable models that should be cited for
completeness: Hesitant Fuzzy Soft Set and Its Applications in Multicriteria
Decision Making. Journal of Applied Mathematics Volume 2014, Article ID
643785.
Authors: We have arranged the text as suggested by reviewer and add hes-
itant fuzzy soft set in the revised manuscript.
Reviewer: Example 2: the parametric PFS describes the view of an expert
on what?
Authors: We have given the description in page 9 (lines 216-218) in the
revised manuscript.
Typos and minor issues:
Reviewer: Line 26: delete one of the two the the
Authors: I have deleted one “the” in page 1 (line 26) of the revised manuscript.
Reviewer: Line 150: scrutinizes.
Authors: I rewrite the paragraph in page 8 (lines 196-205) of the revised
manuscript.
Reviewer: Line 157: we call . soft set.
Authors: Corrected as suggested by reviewer in page 9 (lines 209-211) of
the revised manuscript.
Reviewer: Definition 5: Then their containment .
Authors: Corrected as suggested by reviewer in page 4 (lines 108) of the
revised manuscript.
Reviewer: Definition 6: we should know what a picture fuzzy value is be-
fore using it. This could be done in Definition 4 easily.
Authors: Corrected as suggested by reviewer in Definition 4, page 3 (lines
104-106) of the revised manuscript.
Reviewer: Definition 10: we should know what PF(X) is. Surely it means
the set of all picture fuzzy sets on X, but I think that this is not defined
anywhere.
Authors: I have defined PF(X) in Definition 10, page 5 (lines 135-136) of
the revised manuscript.
Editor Comments to the Author: (There are no comments.)
Moreover, we would like to thank the referees for reading this paper carefully, providing valuable suggestions and comments.
Sincerely yours,
Poom Kumam

Round 2
Reviewer 2 Report
Thank you for your efforts to improve the manuscript.I have no further comments on this version.